# LiteReality: Graphics-Ready 3D Scene Reconstruction from RGB-D Scans

Zhening Huang[1]    Xiaoyang Wu[2]    Fangcheng Zhong[1]
Hengshuang Zhao[2]    Matthias Nießner[3]    Joan Lasenby[1]

[1]University of Cambridge    [2]The University of Hong Kong    [3]Technical University of Munich
https://litereality.github.io

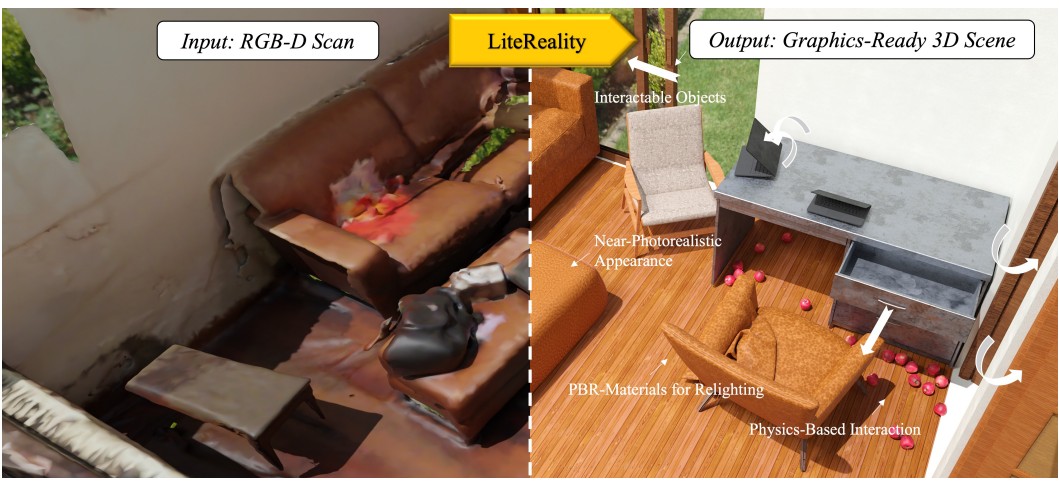

**Figure 1: From RGB-D Scan to Graphics-Ready 3D Scene.** LiteReality reconstructs compact, realistic 3D environments from real-world RGB-D scans, featuring near-photorealistic appearance, articulated geometry, and physically based rendering (PBR) materials—providing assets that can be easily integrated into simulation or rendering pipeline.

## Abstract

We propose LiteReality, a novel pipeline that converts RGB-D scans of indoor environments into compact, realistic, and interactive 3D virtual replicas. LiteReality not only reconstructs scenes that visually resemble reality but also supports key features essential for graphics pipelines—such as object individuality, articulation, high-quality physically based rendering materials. At its core, LiteReality first performs scene understanding and parses the results into a coherent 3D layout and objects, with the help of a structured scene graph. It then reconstructs the scene by retrieving the most visually similar 3D artist-crafted models from a curated asset database. Later, the Material Painting module enhances the realism of retrieved objects by recovering high-quality, spatially varying materials. Finally, the reconstructed scene is integrated into a simulation engine with basic physical properties applied to enable interactive behavior. The resulting scenes are compact, editable, and fully compatible with standard graphics pipelines, making them suitable for applications in AR/VR, gaming, robotics, and digital twins. In addition, LiteReality introduces a training-free object retrieval module that achieves state-of-the-art similarity performance, as benchmarked on the Scan2CAD dataset, along with a robust Material Painting module capable of transferring appearances from images of any style to 3D assets—even in the presence of severe misalignment, occlusion, and poor lighting. We demonstrate the effectiveness of LiteReality on both real-life scans and public datasets.

39th Conference on Neural Information Processing Systems (NeurIPS 2025).

# 1 Introduction

Creating digital replicas of real-world environments remains a central challenge in machine perception, computer vision, and graphics. Although differentiable rendering techniques have achieved remarkable photorealism in 3D reconstruction [41, 22], their outputs often resemble "3D photographs": they capture geometry and texture but lack true interactivity and structure. A truly effective digital replica must go beyond visual fidelity to offer semantically rich, simulation-ready, interactive environments. Such representations empower users to interact with, manipulate, and simulate complex scenes—capabilities essential for robotics, virtual and augmented reality, and embodied AI.

Compared to traditional 3D reconstruction techniques—such as Structure-from-Motion (SfM) [49, 56], Gaussian Splatting [22], and Neural Radiance Fields (NeRF) [41, 7]—we introduce the concept of **graphics-ready reconstruction**. This form of reconstruction goes beyond visual fidelity by incorporating additional structural and functional information to ensure compatibility with graphics pipelines and support downstream tasks such as interactive simulation. To achieve this, several key criteria must be satisfied. First, the scene should be *object-centric*, treating each object as an individual entity to enable physically plausible placement and manipulation. Second, *object functionality* must be modeled, allowing for interactive behaviors such as opening doors, drawers, or appliances. Third, the scene should use *PBR materials* to ensure photorealistic appearance under varying lighting conditions. Lastly, the *integration of physical properties*—including mass, gravity, and collision dynamics—is essential to support realistic interactions grounded in real-world physics. By meeting these requirements, we move beyond passive, geometry-focused reconstructions toward truly graphics-ready environments that combine visual realism with functional structure.

Several research efforts have addressed different aspects of this broader challenge. First, in the simulation domain, recent work study how to create interactive virtual environments from text descriptions [61], room layouts [11], large-scale procedural generation pipelines [12], and even from single images [9]. However, these approaches often prioritize diversity over realism, aiming to generate training data for embodied AI rather than faithfully reconstructing digital replicas of real-world scenes. Other methods pursue realism through compact, abstract representations—for instance, using CAD retrieval and alignment to represent scenes [21, 17, 20, 26, 25] or estimating object textures and materials from images [63, 59, 36]. While promising, these approaches often overlook the importance of integrating components into a coherent, functioning system and tend to neglect the difficulties posed by real-world indoor scans—such as clutter, severe occlusions, and poor lighting. As a result, existing systems frequently struggle to produce faithful reconstructions from in-the-wild scans, limiting their applicability beyond curated or controlled environments.

To this end, we introduce LiteReality, a method that automatically transforms RGB-D scans into realistic, interactive indoor environments (see Figure 1). LiteReality is designed to handle the diverse challenges of real-life scans. At its core, LiteReality consists of four main stages: 1. Scene Perception and Parsing: Objects and room layouts are extracted from RGB-D scans using off-the-shelf methods [53, 28] and represented in a compact scene graph. This graph enforces spatial constraints and refines noisy input into a structured, physically plausible layout. 2. Object Reconstruction: Artist-designed 3D models that best match real-world objects are retrieved from a curated asset database. Though not perfectly aligned geometrically, these models provide high-fidelity geometry and support interactive functionality, including articulated components. 3. Material Painting: A robust material estimation and optimization framework assigns high-quality PBR materials to the retrieved 3D models, based on multi-view RGB images. 4. Procedural Reconstruction: The final scene is assembled by constructing room structures, assembling doors and chairs, placing objects, and assigning basic physical properties and interaction logic—resulting in a lightweight, graphics-ready environment. We evaluate LiteReality on the ScanNet dataset [8] and on real-world indoor scans captured with an iPhone. Across diverse environments, our method consistently converts noisy, incomplete scans into compact, and realistic digital replicas. Our main contributions are as follows:

- **LiteReality framework:** To the best of our knowledge, LiteReality is the first system capable of reconstructing room-level RGB-D scans into compact and realistic CAD representations that feature high-quality PBR materials, providing a strong foundation for downstream applications such as rendering, simulation, and virtual interaction. The system generalizes well across a wide range of real-world scenes.
- **Training-free retrieval pipeline**: LiteReality includes a training-free object retrieval system that achieves state-of-the-art (SOTA) similarity performance on the Scan2CAD benchmark.

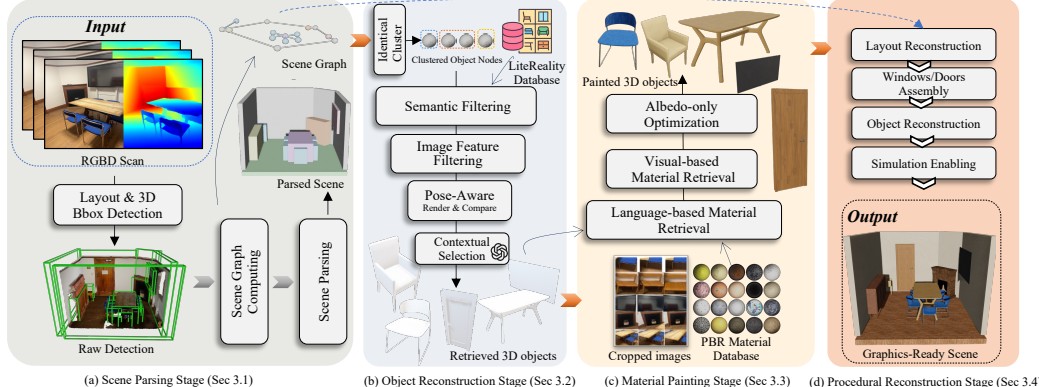

(a) Scene Parsing Stage (Sec 3.1)  (b) Object Reconstruction Stage (Sec 3.2)  (c) Material Painting Stage (Sec 3.3)  (d) Procedural Reconstruction Stage (Sec 3.4)

Figure 2: **Pipeline of LiteReality.** Given input RGB-D scans, the process begins with scene perception and parsing, where room layouts and 3D object bounding boxes are detected and organized into a structured, physically plausible arrangement using a scene graph. In the object reconstruction stage, identical clustering first identifies repeated objects, followed by a hierarchical retrieval procedure that matches 3D models from the LiteReality database. The material painting stage retrieves and optimizes PBR materials by referencing the observed images. Finally, the procedural reconstruction stage assembles all components into a graphics-ready environment featuring realistic appearance and seamless integration with standard graphics pipelines.

- **Robust material painting**: We introduce a novel material-painting approach that reliably transfers high-fidelity PBR materials onto 3D models—even in the presence of shape misalignment, challenging lighting, and diverse image styles. Quantitative evaluation shows SOTA performance.

## 2  Related Work

**Graphics-Ready Environments.**    While industries such as gaming, AR/VR, film production, and physical simulation rely heavily on traditional computer graphics pipelines to create 3D environments, most state-of-the-art 3D reconstruction methods [41, 49, 22] are not readily compatible with these pipelines. This incompatibility poses a barrier to their practical adoption in real-world applications. In the research domain, simulation environments designed for embodied AI training are the closest counterparts to graphics-ready environments, as they emphasize both interactivity and realism. Significant efforts have been made to develop such simulation platforms. AI2-THOR [24], RoboTHOR [10], OmniGibson [30], Habitat 3.0 [48], ThreeDWorld [15], and VirtualHome [47] are notable examples. Despite their capabilities, these platforms often rely on manually crafted scenes, which limits scalability. Several works have proposed scalable scene generation methods. ProcTHOR [12] extends AI2-THOR with procedurally generated environments; Holodeck [61] creates scenes from natural language inputs; Phone2Proc [11], generates room layouts by placing semantically similar objects; and Digital Cousin [9] produces diverse simulated environments from single images. More recent contributions in this area include Scenethesis [35], PAT3D [33] and VIGA [64]. While these approaches improve scalability and diversity, they primarily focus on synthesizing virtual scenes for robotic training rather than faithfully reconstructing real-world environments. This highlights a key gap: building graphics-ready scenes that combine realistic appearance with physical interactivity. A concurrent work, Metascenes [65], is also working in a similar direction by proposing a large-scale dataset that contains a large number of simulatable scenes constructed from richly annotated ScanNet scans for embodied AI benchmarks. LiteReality, by contrast, is presented as a general-purpose tool for converting real-life scans to graphics-ready scenes.

**Object-Centric Reconstructions.**    Joint 3D scene understanding and reconstruction [43, 19, 37, 32] is essential for building object-centric scene representations. Recent advances in 3D scene understanding—particularly in oriented bounding box detection [28, 52] and room layout estimation [5, 34]—have made this direction increasingly promising. An emerging line of work focuses on reconstructing object-centric 3D scenes from a single RGB image [62, 59, 9]. To model individual objects, some methods perform mesh reconstruction from multi-view RGB images [43, 37], while others leverage 3D generative models [62]. Despite rapid and ongoing progress in these domains,

current approaches often suffer from instability or produce low-quality results, limiting their applicability in high-fidelity, simulation-ready applications. An alternative is to retrieve artist-created CAD models, enabling clean, lightweight, and visually coherent scene reconstruction—commonly used in simulation environments. Also, these models often include articulated variants, making them valuable for interactive and setups, though sometimes at the expense of realism. A key direction in this area is CAD model retrieval and alignment for scene reconstruction, as demonstrated by Scan2CAD [3] and many follow-up works [21, 17, 20, 26, 25, 31, 39, 54, 38, 27, 5, 42, 4, 18, 1]. Note that these methods primarily emphasize alignment accuracy post-retrieval, often assuming that retrieving any object from the correct category is sufficient. While some works aim to improve retrieval similarity [1, 2, 55], they typically require accurate instance segmentation—a challenging prerequisite in real-world scan data. In contrast, our retrieval method is explicitly designed to address the challenges posed by real-world scans.

**Materiel Estimation.**  Physically-based rendering (PBR) materials are crucial for achieving photo-realism in advanced applications. Estimating PBR materials typically requires recovering spatially varying bidirectional reflectance distribution function (SVBRDF) parameters from RGB images—an inherently ill-posed problem due to the complex interaction of lighting, geometry, and reflectance. A common approach involves retrieval followed by optimization, where materials are initially retrieved based on visual similarity and then refined via differentiable rendering [50, 29]. For instance, PSDR-Room [59] and PhotoScene [63] optimize procedural PBR parameters from image observations using differentiable rendering. Similarly, MAPA [66] carries out part-level segmentation and style transfer, while Material Palette [36] uses a learning-based framework to estimate plausible materials from segmented object views. Although effective in clearly visible and well-cropped scenarios, these methods struggle in cluttered, room-scale environments. Estimation quality often degrades due to occlusion, inaccurate segmentation, low-resolution inputs, and complex lighting conditions. Additionally, per-object differentiable rendering makes these pipelines computationally expensive, limiting their scalability to scenes with many objects. In LiteReality, we introduce a self-contained and robust method tailored for room-level material estimation in messy, real-world scans. Our method generalizes well under occlusion and partial visibility, and remains reliable across diverse lighting conditions.

# 3  Methodology

**LiteReality** is a framework that automatically converts indoor RGB-D scans into Graphics-ready scenes. It begins by performing scene perception to extract the key information of layout and objects. The scene is then procedurally constructed in a physics engine to enable interactivity and simulation capabilities. At its core, the pipeline consists of four key stages: *scene perception and parsing*, *object reconstruction*, *material painting*, and *procedural reconstruction*, as shown in Figure 2. These stages work together to automatically assemble a complete, graphics-ready environment. We highlight the key features and design of each stage in the following sections.

## 3.1  Scene Perception and Parsing

**Scene Understanding.**  LiteReality takes RGB-D images of a room as input. In the scene perception and parsing stage, it extracts key spatial information necessary for producing graphics-ready reconstructions. Among various 3D scene understanding approaches, we identified two key tasks—room layout estimation and oriented bounding box detection—as particularly effective due to their robustness and scalability for generating structured 3D environments. Specifically, for real-life scans, we used Apple's RoomPlan [53] with an iPhone to capture layouts and oriented bounding boxes (O-Bbox) interactively. There are also many open-source alternatives for layout and O-Bbox detection [40, 6, 52] that can be used. Raw detection results often exhibit noise, such as collisions and misalignments, which can result in issues like overlapping objects, floating items, and discontinuous walls when constructing a 3D world. These problems break physical plausibility and break realism. To address this, we propose a *scene graph representation* that organizes the scene by explicitly representing spatial and appearance relationships between objects and their surroundings. This structured representation is essential for guiding later constrained collision resolution and ensuring consistency in later stages of the pipeline.

**Scene Graph Representation and Parsing.** In the scene graph, nodes represent key elements in the scene, such as walls, windows, doors, and detected objects. Each node is associated with several attributes: 1) **spatial properties**, including center location $\mathbf{C} \in \mathbb{R}^3$ (where $\mathbf{C} = (x, y, z)$), dimensions $\mathbf{D} \in \mathbb{R}^3$ (where $\mathbf{D} = (w, h, l)$), and orientation $\boldsymbol{\theta} \in \mathbb{R}^3$ (three rotation angles around the $x$, $y$, and $z$ axes); and 2) **appearance properties**, represented by the top-$k$ most visible cropped images of the object, denoted as a set $\mathcal{I} = \{I_1, I_2, \ldots, I_k\}$. These attributes are essential for material estimation, model selection, and layout optimization. Edges between nodes define key spatial relationships and, once constructed, can impose constraints during the spatial arrangement process, ensuring a physically plausible scene. We define four key spatial relationships: 1) **attached to walls**, for objects physically connected to walls (e.g., shelves, cabinets); 2) **on top of**, for objects placed on top of others (e.g., a vase on a table); 3) **table-chair pair**, grouping tables and chairs in typical arrangements; and 4) **connecting to**, where objects are next to each other and touched. These relationships provide essential constraints for maintaining consistent object placement, preventing collisions, and improving overall scene realism. To compute the edges of the scene graph, the construction process begins by parsing walls into connected components, forming closed polygonal regions that delineate individual rooms. Noisy wall detections are aligned by snapping them to a grid, ensuring well-defined room boundaries. Spatial relationships between objects are inferred based on their **center location** and **orientation**, with predefined rules governing each type of relationship. We resolve object collisions iteratively using a **Constraint-Based Collision Resolution** approach. For each intersecting pair, virtual forces are applied along collision vectors to separate them, while adhering to spatial constraints from the scene graph. For example, wall-attached objects are restricted to wall-aligned movement, and table-chair pairs retain their relative positions. This process continues until collisions are resolved or a maximum iteration count is reached, ensuring a realistic, physically plausible layout. More details can be found in the supplementary materials.

### 3.2 Object Reconstruction Stage

In the age of ever-expanding 3D asset datasets, it is crucial to develop a robust and scalable retrieval mechanism. Previous approaches [27, 4, 3] often rely on pair-wise pre-trained models for retrieval, but these are limited to specific datasets. We believe it is important to adopt a *training-free* approach, as it would be more adaptable and scalable when the dataset expands. We adopt a hierarchical retrieval approach that leverages multiple information modalities—semantic labels, visual features, and camera pose—to progressively refine the search space and improve accuracy. The retrieval begins with a semantic filtering stage, where objects are categorized by subcategories (e.g., two-seat sofas, bar chairs) to exclude irrelevant candidates. This is followed by an image-based retrieval stage, where cropped 2D views of the input object are compared against pre-rendered images of candidate objects using a pretrained feature encoder (DINOv2[45]). The top 10 visually similar matches are selected. In the subsequent pose-aware rendering and comparison step, these candidates are placed into the scene according to the detected pose and rendered from the same camera angles. The resulting views are then cropped and re-encoded for visual feature extraction, further narrowing the selection to the top four candidates. Finally, a contextual selection step employs a language model to assess high-level attributes such as style, proportion, and visual coherence, yielding the best-matched object. This structured, multi-stage pipeline ensures scalable and context-aware retrieval across large datasets. To group identical or highly similar objects, we cluster items within each subcategory using DINOv2 features extracted from cropped images. These features, which capture shape and structure, are averaged per object and clustered via KMeans, with the number of clusters selected by maximizing the silhouette score. Since DINOv2 is color-invariant, we further subdivide clusters by their dominant color to form fine-grained, visually and stylistically consistent groups for joint retrieval.

### 3.3 Material Painting Stage

One of the primary challenges in the pipeline lies in effective and accurate material recovery. While prior approaches [59, 63, 36] have made notable progress using differentiable rendering or prediction-based techniques, they struggle to scale effectively to real-world, noisy environments. This limitation arises from three key factors. First, these methods rely on precise alignment between material segments and cropped image regions. In practice, due to geometric misalignment of objects, their approaches to obtaining reliable crop mappings often fail, resulting in noisy or inconsistent references. Second, poor lighting conditions frequently lead to dark or visually ambiguous crops which, even if correctly mapped to material segments, still result in degraded material initialization. Finally, procedural, graph-based differentiable material rendering introduces significant computational overhead,

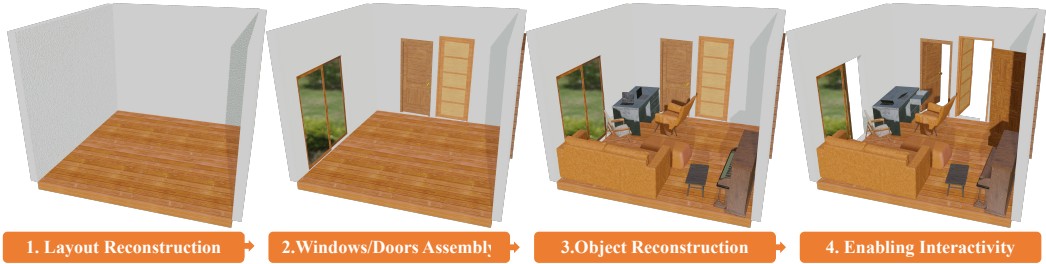

| 1. Layout Reconstruction | 2.Windows/Doors Assembly | 3.Object Reconstruction | 4. Enabling Interactivity |

Figure 3: **Procedural Reconstruction Stage.** This stage progressively rebuilds rooms by reconstructing layouts, assembling doooors and windows, placing objects, and enabling interactive attributes.

limiting the ability to process large-scale scenes efficiently. In the **Material Painting Stage**, we address prior limitations with a compact pipeline that includes automatic crop mapping, multi-step material selection, and lightweight albedo-only optimization. For each retrieved 3D asset, predefined material groups are assigned materials inferred from captured images.

**Auto-crop Mapping.** The input to this stage includes multiple cropped image patches and corresponding 3D models. We begin by applying SAM [23]-guided segmentation to extract meaningful patches from the images. Specifically, we compute the largest rectangular region within each mask and filter out invalid segments using Grounding DINO detection. Priority is given to smooth, clean regions and larger areas, from which we select the top-*k* ranked patches. To associate 3D material segments with these image patches, we generate informative visual prompts and employ a Multi-Modal Large Language Model (MLLM) [44] for semantic mapping.

**Semantic and Visual Guided Material Search.** To initialize high-quality materials, we combine language-guided and vision-based retrieval from a material database. First, inspired by Make-It-Real [13], we use multi-step prompting to refine material category predictions and select the top 10 candidates. Then, we extract CLIP embeddings from cropped multi-view patches and compare them with embeddings of the selected materials. Finally, GPT-4 evaluates the visual compatibility of albedo maps with reference patches, ensuring semantic and visual alignment. This multi-stage process produces reliable PBR material initialization for each 3D segment.

**Albedo-Only Optimization.** With initial material assignments in place, further refinement is possible using procedural graph-based methods like MaTCH [50] or PSDR-Room [59], but these incur high computational costs. Instead, we apply a lightweight, albedo-only adjustment in the CIE LAB color space. Rather than re-optimizing full PBR parameters, we shift the global color distribution of the albedo to better match cropped image patches—preserving high-frequency details while correcting hue and brightness. Formally, let $\mathbf{S}(p) \in \mathbb{R}^3$ be the LAB vector of the source albedo at pixel $p$, $\mathbf{T} \in \mathbb{R}^3$ the target LAB vector, and $P$ the set of pixels in the albedo map. The adjusted color is:

$$\mathbf{S}'(p) = \mathbf{S}(p) + \left( \mathbf{T} - \frac{1}{|P|} \sum_{q \in P} \mathbf{S}(q) \right).$$

This shift aligns the mean color of the albedo with the target while preserving texture details. To infer $\mathbf{T}$, we query an MLLM for part-specific RGB values across views, convert the consensus to LAB, and apply the shift. This efficient post-processing balances visual fidelity and scalability.

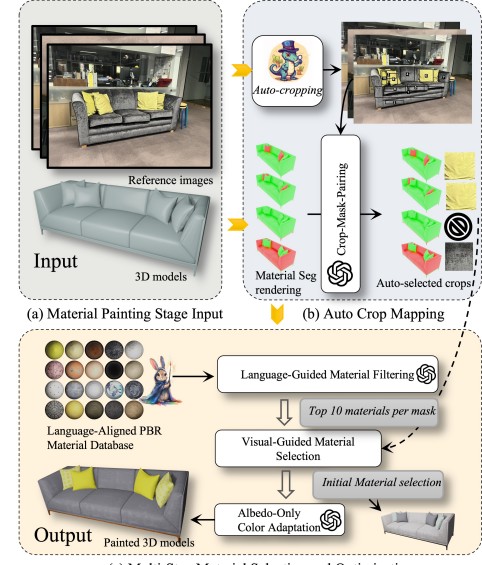

(a) Material Painting Stage Input

(b) Auto Crop Mapping

(c) Multi-Step Material Selection and Optimization

Figure 4: **Materials Painting Stage Pipeline**. Given a 3D model and reference images, this stage recovers realistic PBR materials that enhance visual realism for the 3D object.

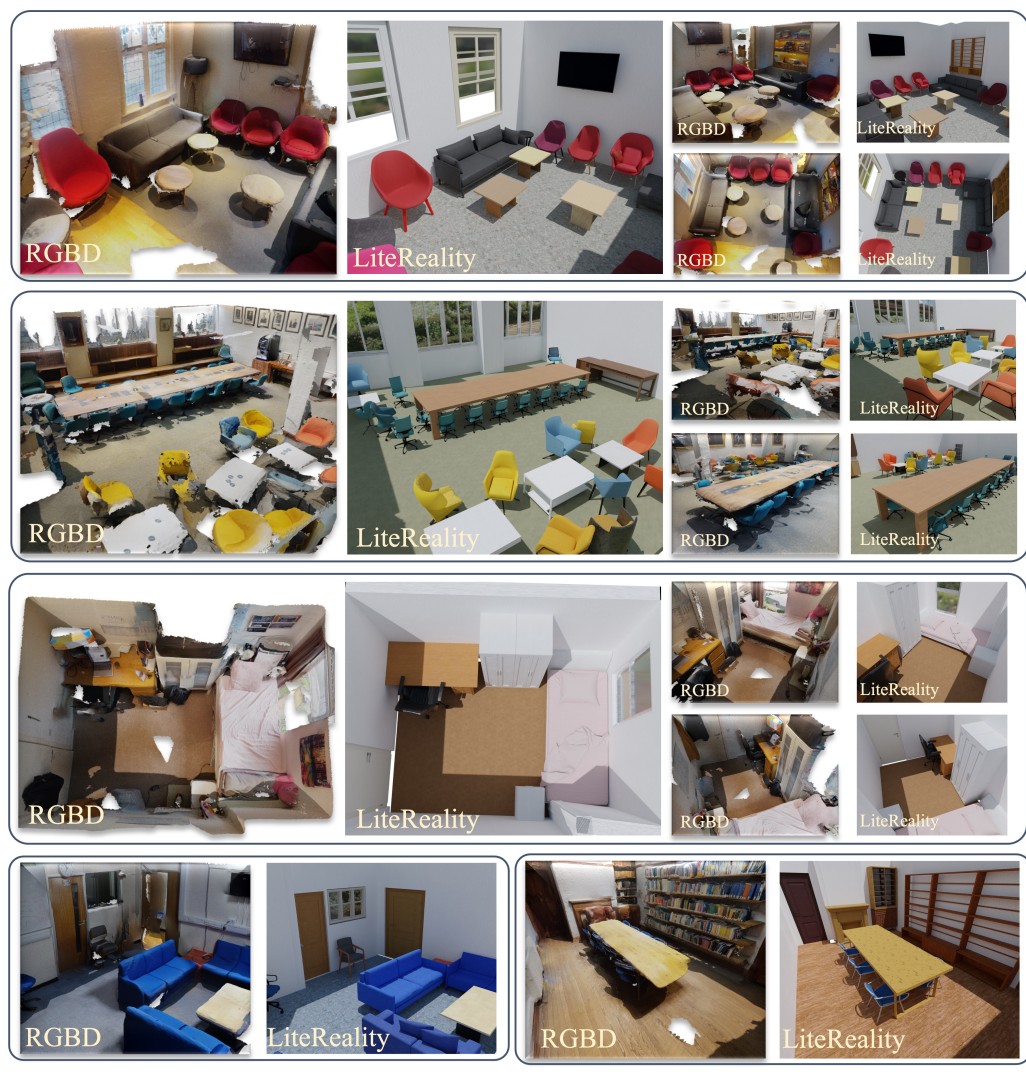

Figure 5: **Graphics-Ready Reconstruction by LiteReality.** Noisy indoor scans are converted into compact, realistic scenes with full PBR materials for all objects, ready for downstream graphics tasks.

## 3.4 Procedural Reconstruction Stage in Blender

Finally, we use a predefined procedural pipeline to reconstruct the scene. The process begins with constructing the walls, followed by assembling windows and doors, and finally placing objects within the room. To support physics-based interaction, we assign rigid-body properties to each object and use their mesh geometry as collision boundaries. Large structural elements, such as walls, are marked as passive rigid bodies, while movable items are set as active rigid bodies. Estimating object mass is crucial for realistic simulation. To achieve this, we crop images of each object and input them into a Multimodal Large Language Model (MLLM), which predicts their mass—later used to simulate realistic rigid-body dynamics. The entire procedural reconstruction process is carried out within Blender.

## 4 Experiments and Results

We evaluate LiteReality both end-to-end and per stage. Since no prior work directly tackles this task, we propose three benchmarks: retrieval similarity, object-centric PBR material estimation, and full-scene graphics-ready reconstruction. These enable quantitative comparisons with prior work and provide a foundation for future research. We also demonstrate advanced applications of our outputs in the supplementary material.

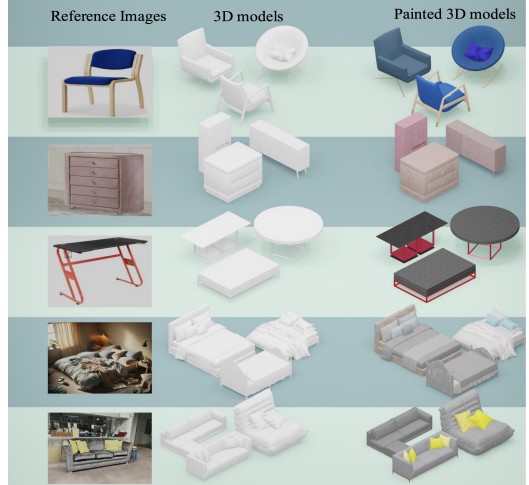
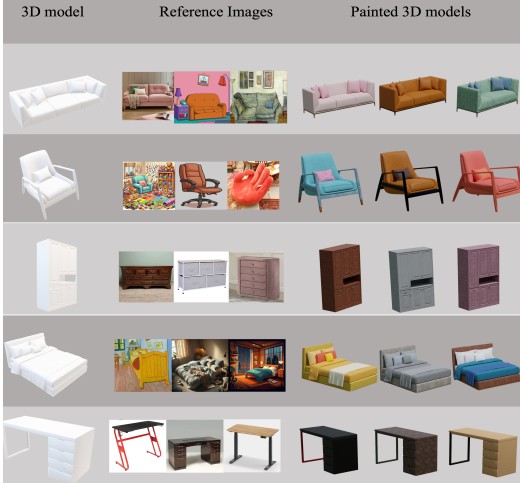

Figure 6: **Material Painting Stage Demo**: Single reference image applied to multiple 3D models. Close-to-reality PBR materials are recovered, even with sever geometries misalignments.

Figure 7: **Material Painting Stage Demo**: A 3D model is painted using diverse reference images. Realistic PBR materials are recovered, even under challenging conditions like cartoons or drawings.

**LiteReality Database.** LiteReality database is structured to align with Apple's RoomPlan, supporting real-world scene reconstruction across 17 semantic classes. We organized our dataset accordingly, ensuring each category—and relevant subcategories—has a corresponding retrieval database. To build this database, we curated assets primarily from 3D-Future [14] and AI2-THOR [24]. 3D-Future offers diverse indoor furniture, while AI2-THOR, though smaller, provides detailed articulation data for interactive scenes. For underrepresented categories, we supplemented assets from Sketchfab [51] under public-access licenses. As of May 2025, our retrieval database includes 5,283 assets; full statistics are available in the supplementary materials.

**Retrieval Similarity Benchmarking.** To evaluate the performance of proposed object retrieval method, we conduct experiments on the ScanNet dataset [8]. Since Scan2CAD[3] provides hand-picked ShapeNet[57] models as ground-truth annotations, we perform retrieval over the full ShapeNet database. Following [2, 16, 55], we measure retrieval accuracy using the $L_1$ Chamfer Distance between the retrieved model and the ground-truth. Unlike Scan2CAD, which searches within a per-scene CAD pool of roughly 50 candidates, we retrieve from the entire ShapeNet repository, where each category contains 300–3 000 models. Although similar evaluations have been conducted before, direct comparison is hindered by sparse implementation details in prior work. We provide a complete description of our evaluation protocol in the supplementary material to facilitate reproducibility and future comparisons. Experiments use the ScanNet validation scenes.

**Object-Centric Material Recovering Benchmarking** BRDF estimation of retrieved objects based on scanned images brings the appearance of real-world objects into the digital space. To evaluate the performance of our proposed method, we established a benchmark using five indoor scenes captured with an *iPhone 13 Pro Max* running the 3D Scanner App in RoomPlan mode. The scenes include a meeting room, a common area, a bedroom and two study rooms, contains a total of 111 objects across eight categories. For each object, we selected the four most visible frames and tightly cropped each to isolate the object, highlighting real-world scanning challenges such as occlusion, poor lighting, and misalignment. We compare our material painting pipeline against four baselines: (1) **PhotoShape**[46], a material selection algorithm; (2) **Make-It-Real**[13], which use language model to retrieve materials; (3) our **Visual+Language Search** method; and (4) **Make-It-Real with our Albedo-Only Optimization**. To enable benchmarking, we also created ground-truth CAD models for the scenes. PBR materials estimated by each method were applied to the models, which are placed at their original poses, and rendered with global illumination using HDR environment maps. Evaluation was based on perceptual differences between the cropped scan images and the corresponding rendered views. More details are presented in the supplementary materials.

**Graphics-Ready Reconstruction Benchmarking.** To assess end-to-end reconstruction quality, we introduce a benchmark for graphics-ready reconstruction. Only scenes that are object-centric

Table 1: **Quantitative Evaluation of Retrieval Similarity.** We compute the two-way Chamfer Distance between the normalized GT CAD models annotated in Scan2CAD and our retrieved results. Evaluation is performed on the ScanNet validation set. Our retrieval method consistently outperforms prior approaches, achieving the most accurate shape matches across multiple object categories.

| Method | avg/CAD ↓ | avg/class ↓ | bath ↓ | bkslf ↓ | cab ↓ | chr ↓ | disp ↓ | sfa ↓ | tbl ↓ | bin ↓ |
|---|---|---|---|---|---|---|---|---|---|---|
| MSCD [55] | 0.1103 | 0.1188 | 0.1215 | 0.1114 | 0.0931 | 0.1019 | 0.1423 | 0.1071 | 0.1224 | 0.1119 |
| Digital Cousin [9] | 0.1411 | 0.1246 | 0.1439 | 0.1166 | 0.1105 | 0.1363 | 0.1762 | 0.1083 | 0.1832 | 0.1098 |
| ScanNotate [1] | 0.1042 | 0.1110 | 0.1161 | 0.0870 | 0.0908 | 0.0995 | 0.1376 | 0.1046 | 0.1183 | 0.0921 |
| **LiteReality** | **0.0986** | **0.1067** | **0.1109** | **0.0845** | **0.0859** | **0.0943** | **0.1309** | **0.0951** | **0.1099** | **0.0901** |

Table 2: **Object-Centric PBR Material Estimation Comparison**. Evaluated on 110 objects across five real-world scanned scenes, LiteReality method demonstrates strong capabilities.

| Methods | RMSE ↓ | SSIM ↑ | LPIPS ↓ |
|---|---|---|---|
| MIR [13] | 0.2377 | 0.3981 | 0.6111 |
| PhotoShape [46] | 0.3225 | 0.2371 | 0.6558 |
| MIR [13] + AO | **0.2156** | 0.4203 | 0.5899 |
| Sem&Vis | 0.2835 | 0.3758 | 0.6362 |
| **LiteReality** | 0.2163 | **0.4353** | **0.5854** |

Table 3: **Perceptual Quality Evaluation for Full Scenes**. Tested by comparing rendered images to captured RGB, LiteReality significantly outperforms prior works across all metrics.

| Methods | RMSE ↓ | SSIM ↑ | LPIPS ↓ |
|---|---|---|---|
| Phone2Proc[11] | 0.3604 | 0.5512 | 0.7338 |
| Digital Cousin [9] | 0.3653 | 0.5531 | 0.7364 |
| DC [9] + Sem&Vis | 0.3226 | 0.5425 | 0.6717 |
| DC [9] + MIR [13] | 0.3046 | 0.5492 | 0.6648 |
| **LiteReality** | **0.2664** | **0.5818** | **0.6522** |

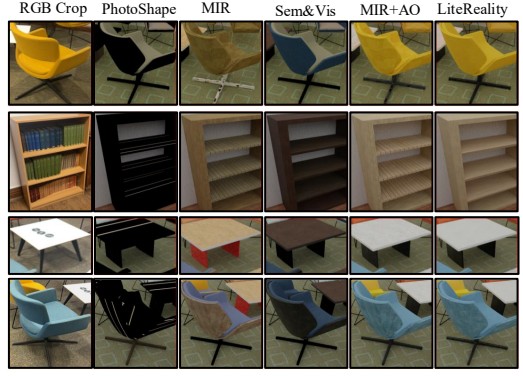

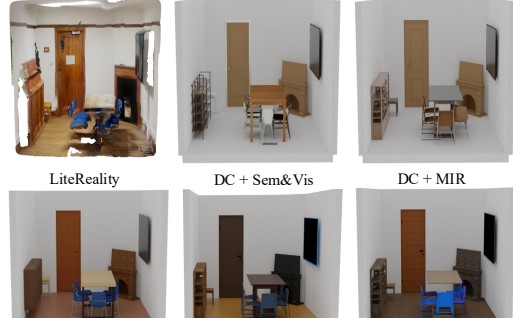

Figure 8: Object-Centric PBR Material Comparison . Note: *object pose are unused in MP pipeline*

Figure 9: Graphics-Ready Reconstruction Comparison. LiteReality showcases best realism.

and realistic (i.e., each object is an individual entity with PBR materials) qualify for graphics-ready reconstruction evaluation. We evaluate four methods in total. First, we reproduce two prior approaches: Phone2Proc [11] and Digital Cousin. Second, we extend the Digital Cousin framework by integrating material selection via (1) the original Make-It-Real (MIR) pipeline and (2) our proposed semantic-and-vision guided scheme. All methods are tested on five diverse scenes captured with an iPhone, each containing hundreds of RGB frames. The results are shown in Table 3.

**Results.** As shown in Table 1, our training-free retrieval method achieves the best similarity matching among all baselines. Table 2 quantifies the effectiveness of our material painting approach, clearly outperforming prior methods. For overall scene reconstruction, LiteReality shows strong overall performance in Table 3, achieving the highest visual quality. Figures 5 and 9 illustrate results across diverse real-world scenes and comparisons with baselines. Figure 8 showcases the superior visual results of our proposed methods in PBR comparisons. We further validate robustness in two challenging scenarios: painting multiple 3D models from a single image (Figure 6) and painting a single model using diverse images (Figure 7), including cartoons, drawing or AI-generated content. In both cases, our method produces complete, visually plausible PBR materials that matched that in the image. Additional applications—relighting, editing, physics, AR/VR, and robotics—are shown in the supplementary material.

**Runtime Analysis** On a single NVIDIA RTX 3090 GPU with 24 GB of memory, the complete reconstruction of a room-scale scene takes between 20 and 60 minutes, depending on scene complexity. For example, a small study or bedroom scene containing around 10–15 objects typically requires about

20 minutes, whereas larger spaces such as meeting rooms or boardrooms with 40–50 objects take up to an hour. Among the individual stages, preprocessing and scene parsing complete within 1–3 minutes, object retrieval requires 2–5 minutes, while the material painting stage dominates the runtime, taking roughly 15–50 minutes depending on the number of objects. Procedural reconstruction and export in Blender add less than two minutes on average.

## 5    Conclusion

**LiteReality** demonstrates the ability to reconstruct realistic and interactive indoor scenes from RGB-D scans. Our results highlight the potential of the proposed pipeline in generating visually compelling environments with physics-based interactions and dynamic lighting. We hope this work will inspire further exploration toward graphics-ready 3D scene reconstruction.

**Limitations.**    The overall reconstruction quality remains closely tied to the accuracy of upstream perception. Errors in object detection and layout estimation can propagate to later stages. In most real-world cases, our input scans are captured using Apple's RoomPlan app, which provides reasonably reliable geometry through interactive scanning. While we propose a comprehensive retrieval pipeline, object retrieval remains inherently challenging—an occasional mismatch can noticeably affect scene realism. Fortunately, due to the object-centric nature of our representation, such errors are easy to correct through local replacement or refinement. Finally, our current constraint-based collision solver may struggle in densely packed or complex arrangements, sometimes leading to interpenetrations and preventing fully physics-valid simulations.

**Future Work.**    Future work could focus on extending object detection and retrieval to include smaller items, thereby enriching scene detail and overall realism. Incorporating learning-based light estimation together with full SVBRDF recovery would enable faithful relighting under novel illumination conditions. Scaling the pipeline from single-room to multi-room and building-level reconstructions would further unlock applications in large-scale visualization, robotics simulation, and digital-twin creation. In addition, optimizing the entire system for real-time or near real-time performance would make it highly practical for interactive applications and on-device deployment.

**Acknowledgment.**    This work was supported by the Girton College Graduate Research Award at the University of Cambridge, the School of Technology, and the EPSRC Centre for Doctoral Training. We gratefully acknowledge the insightful discussions and feedback from Prof. Kwang Moo Yi, Erqun Dong, Weiyang Liu, Prof. Ayush Tewari, and Shangzhan Zhang throughout the course of this work. We also appreciate the kind support of Prof. Angela Dai for providing the voiceover for the demo videos, and we thank Jieying Chen for her help in proofreading the paper.

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

# Appendix

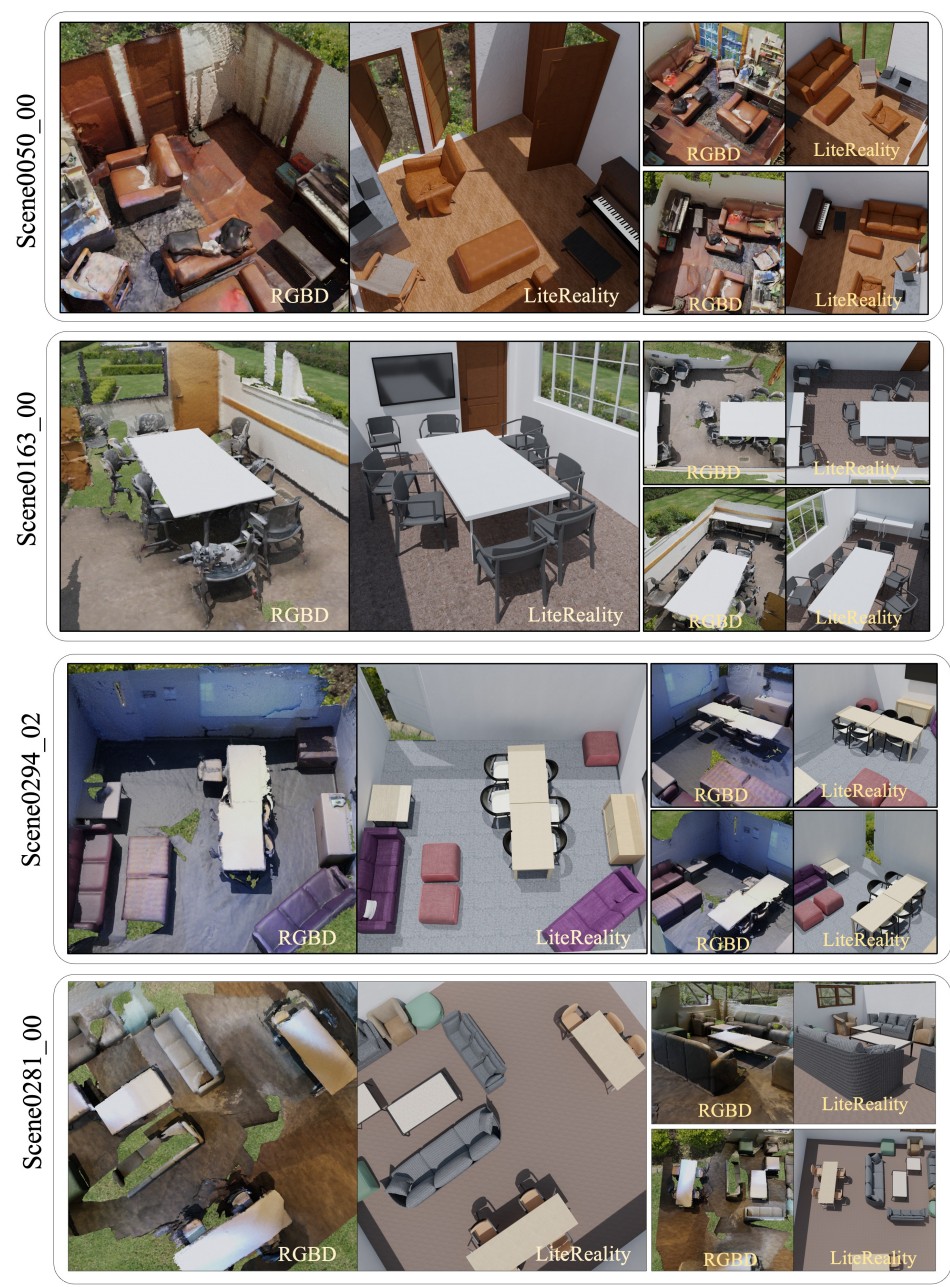

Figure 10: **Graphics-Ready Reconstruction by LiteReality on ScanNet Dataset** Noisy indoor scans are converted into compact, realistic scenes with full PBR materials for all objects, ready for downstream graphics tasks.

# A    Outline

In this supplementary document, we present five key sections that provide additional context and depth to our main paper:

1. **Database Explanation.** We explain the construction of our *LiteReality* database and discuss how it can be expanded in future work.

2. **Benchmark Details.** We describe the construction and presentation of three benchmarks, including the rationale behind their design and implementation details to ensure reproducibility.

3. **Methodology Details.** We provide further description of each stage in our reconstruction pipeline.

4. **Additional Results.** We present further reconstruction results on the ScanNet dataset, including qualitative examples and accompanying supplementary videos.

5. **Application Demonstrations.** We showcase several advanced applications enabled by our Graphics-Ready Reconstruction, demonstrating its versatility in the rendering pipeline, virtual reality, robotics, and scene editing.

Finally, we discuss broader limitations of our current system and outline directions for future work.

# B    LiteReality Database

The overarching goal in designing **LiteReality** is to create a robust system capable of converting real-world scans into *graphics-ready environments* suitable for downstream applications such as VR, robotics, and interactive rendering. Many of our design decisions and technical innovations are driven by the challenge of working with in-the-wild scans. Currently, one of the most reliable solutions for high-quality bounding box and layout detection is Apple's **RoomPlan** system [53], which enables interactive indoor scanning with augmented reality support. Our database is thus specifically tailored to be compatible with this capture system.

The RoomPlan framework recognizes 17 key room-defining furniture categories, including *bathtub, bed, chair, fireplace, oven, refrigerator, sink, sofa, stairs, storage, stove, table, toilet*, and *washer-dryer*, as well as architectural elements like *windows* and *doors*. Accordingly, LiteReality includes these 17 major categories. For broad object types—such as *chairs, tables, storage units*, and *beds*—we further divide them into finer subcategories to better reflect the diversity encountered in real-world settings.

Our object assets are sourced from a combination of public datasets, including **3D-FUTURE** [14], the articulated dataset from **AI2-THOR** [24], and several curated handheld scans from **Sketchfab** [51]. For most models, we provide *material-level segmentation* to support the later Materiel Painting stage. Articulated assets—primarily from AI2-THOR—are carefully processed to maintain sub-mesh structures that accurately represent jointed or movable parts. Note that the number of articulated furniture models remains limited, expanding this subset represents an important direction for future development.

Each 3D model in LiteReality undergoes the following on-boarding steps:

1. **Multi-view rendering:** We generate object renders from multiple viewpoints and extract DINOv2 features from these images, which are stored for use in later retrieval stages.

2. **Orientation normalization:** Object orientations are aligned to be front-facing, ensuring consistent placement within axis-aligned bounding boxes.

3. **Material segmentation rendering:** We extract material segments and generate per-part segmentation renders from multiple viewpoints, then select the most visually representative image for each material segment.

4. **Material segmentation visualization:** We create informative visualizations of material segmentations by stitching together the representative images for each material part. These visualizations are used in later stages for material matching and retrieval, and have been

shown to outperform SOM-based [60] labeling methods when used with MLLM prompts (figure 11).

Figure 12 summarizes the database statistics for each subcategory, showcasing material segmentation availability and examples of articulated models with interactive components.

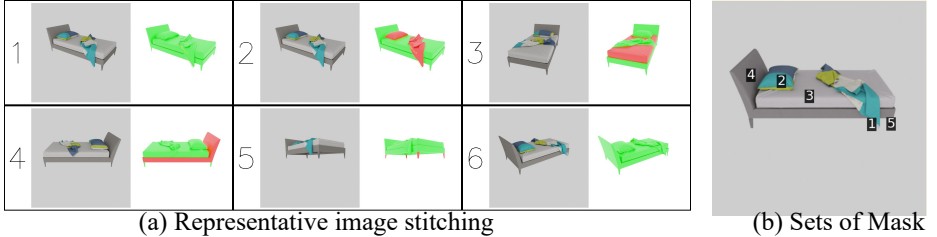

(a) Representative image stitching         (b) Sets of Mask

Figure 11: **Image stitching for materials image prompts in MLLMs.** We tested that image stitching provides more effective visual prompts for MLLMs compared to the commonly used SOM approach.

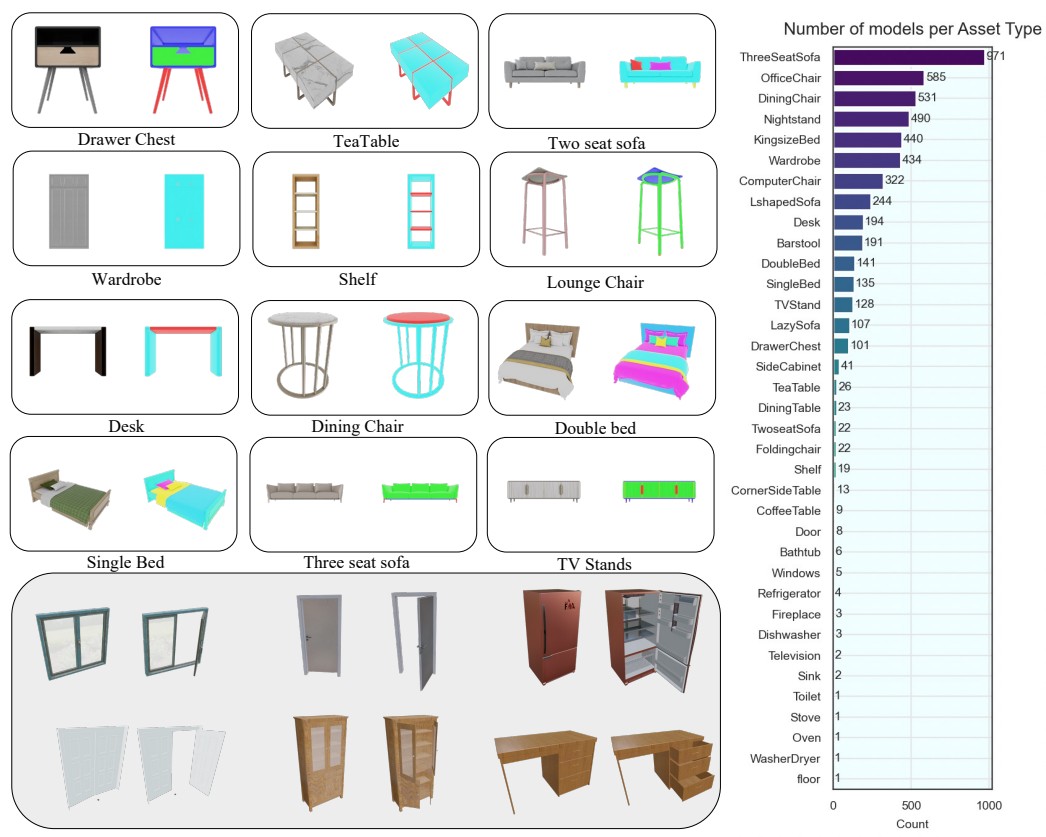

(a) Example 3D models from the LiteReality Database. Each object includes material segmentation, as color-coded on the right (**top**). Some models also feature articulated components, enabling interactive settings (**bottom**).

(b) Models counts per asset categories, constructed in the combination of 3D-Future, and AI-2Thor.

Figure 12: Literality Database: Statistics and Examples

## C    Benchmarking Details

### C.1    Retrieval Similarity Evaluation Protocol

Previous work on CAD model retrieval and alignment for 3D scenes—such as Scan2CAD [3], Vid2CAD [39], and FastCAD [27]—has primarily focused on alignment accuracy. These methods

typically follow the Scan2CAD evaluation protocol, which considers a retrieval successful if the retrieved CAD model belongs to the correct object category, and then evaluates performance based on alignment precision.

In contrast, our pipeline does not perform explicit alignment. Instead, we rely on an off-the-shelf detector to predict 3D oriented bounding boxes for objects, making the Scan2CAD alignment-based protocol inapplicable to our setting.

To evaluate retrieval similarity, we adopt more rigorous and fine-grained metrics proposed in recent literature. Works such as DiffCAD [16], HOC-Search [2], ScanAnnotate [1], Accurate Instance-Level CAD Model Retrieval [55], and FastCAD [27] introduce similarity-based metrics that go beyond categorical correctness. Based on comparisons across these approaches, we adopt the **L1 Chamfer Distance** as our primary metric, measuring the geometric difference between the normalized retrieved CAD model and the normalized ground-truth model (as annotated in Scan2CAD). Our use of this metric is consistent with prior works [2, 16, 55].

Unlike the Scan2CAD protocol, which limits retrieval to a predefined set of CAD candidates, we perform retrieval over the entire ShapeNet category corresponding to each ground-truth object. The number of CAD models per category used in our evaluation is listed in Table 4.

Table 4: Number of CAD model templates in the ShapeNet database for retrieval

| Trash Can | Bathtub | Bookshelf | Cabinet | Chair | Display | Sofa | Table |
|---|---|---|---|---|---|---|---|
| 343 | 477 | 452 | 1571 | 6778 | 1093 | 3173 | 8436 |

Although some earlier works report similarity evaluations, direct comparison remains difficult due to missing details or unavailable implementations. To support reproducibility, we provide a comprehensive description of our evaluation methodology.

**Representative Image Selection**

Images of CAD models play a key role in retrieval pipelines that involve 2D feature extraction, silhouette matching, or depth comparisons. To select representative views for each CAD object, we reproject all 3D points of a scene instance into the image space and choose the top four images with the highest number of visible points—i.e., the least occluded views. For ShapeNet CAD models, we use the pre-rendered multi-view dataset from DISN [58], which contains 32 rendered images per model from different angles.

**Chamfer Distance Computation**

To compute Chamfer Distance, each CAD model is normalized—centered at the origin and scaled to fit within a unit cube ($1 \times 1 \times 1$). We uniformly sample 10,000 surface points from both the retrieved and ground-truth CAD models. The L1 Chamfer Distance is calculated bidirectionally as follows:

$$\text{CD}_{L1}(A, B) = \frac{1}{|A|} \sum_{a \in A} \min_{b \in B} \|a - b\|_1 + \frac{1}{|B|} \sum_{b \in B} \min_{a \in A} \|b - a\|_1 \tag{1}$$

The following three baselines are selected for comparison.

- **SCANnotate** [1]: Utilizes an automatic CAD retrieval pipeline that leverages depth maps, silhouette comparison, and Chamfer Distance via differentiable rendering.

- **Digital Cousin** [9]: Matches CAD models using 2D image features from multiple views to rank visually and semantically similar assets.

- **MSCD** [55]: Employs a two-step approach: initial retrieval using 3D feature descriptors, followed by geometry-based re-ranking using a Modified Single-direction Chamfer Distance (MSCD), which improves robustness to noise and better captures shape similarity in large databases.

We conduct the evaluation on all scenes from the ScanNet[8] validation set.

## C.2 Object-Centric Material Recovery Benchmark

To evaluate object-centric material recovery in realistic conditions, we introduce a benchmark designed to reflect the challenges of real-world RGBD capture. Unlike global scene-level material estimation, our focus on object-level recovery aligns naturally with the object decomposition pipeline used in graphics, and enables per-object physically based rendering (PBR) material assignment—an essential representation for high-fidelity simulation and interactive graphics applications.

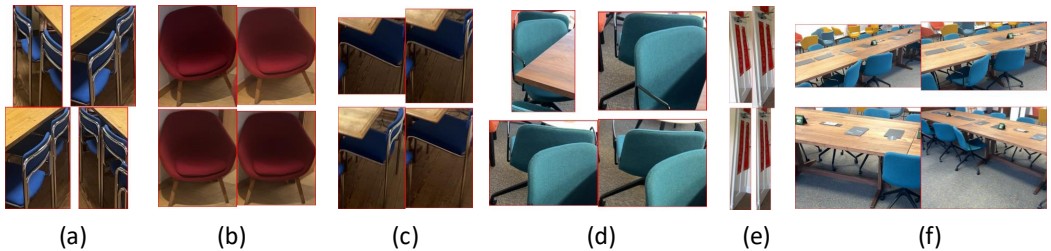

(a)       (b)       (c)       (d)       (e)       (f)

Figure 13: Challenging real-world conditions, including occlusion, poor lighting, skewed angles, and numerous distractors.

We constructed this benchmark using five indoor scenes captured with an iPhone 13 Pro Max via the 3D Scanner App in RoomPlan mode. These scenes cover a range of real-world environments, including a meeting room, a common area, a bedroom, and two study rooms, and contain a total of 111 manually curated objects spanning eight categories.

For each object, we identified and cropped the four most visually representative frames from the RGBD scan. These images often reflect real-world challenges such as occlusion, suboptimal lighting, and partial visibility—conditions under which we aim to test the robustness of various material recovery approaches. Some exampel crops are list in Figure 13. To benchmark performance, we compare our material painting pipeline against four baselines:

1. PhotoShape [46] – a material selection method using appearance-based retrieval;

2. Make-It-Real [13] – a language-guided material retrieval pipeline;

3. Our Visual+Language Search – combining visual similarity and textual prompts;

4. Make-It-Real + Albedo-Only Optimization – a variant incorporating our optimization stage.

Ground-truth CAD models are provided for all scenes. Estimated PBR materials from each method are applied to the CAD models and rendered at their original 3D poses using global illumination with HDR environment maps. For evaluation, we compare the rendered images of cropped objects against the corresponding cropped RGB images using perceptual metrics, as shown in Figure 14. To ensure fair evaluation, we manually selected retrieval results and object categories, avoiding retrieval-induced biases. Note that while we use known object poses for rendering and evaluation, these poses are not used during material recovery to preserve generalization to in-the-wild scenarios. We will release the scanned scenes, selected objects, cropped image sets, and evaluation scripts for reproducibility and support future research in object-centric material recovery.

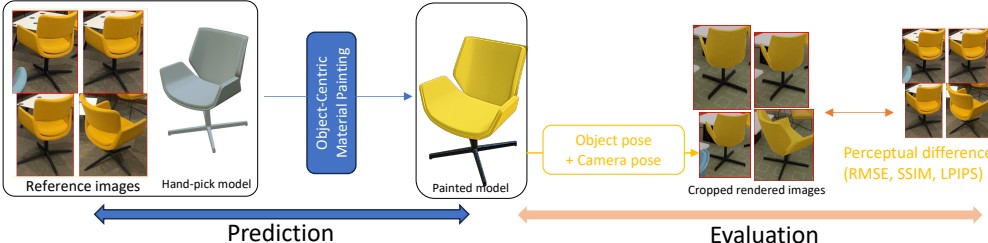

Figure 14: Evaluation protocol used for assessing object-centric material painting performance.

## C.3 Graphics-Ready Reconstruction Benchmark

We use five indoor scenes captured using an iPhone 13 Pro Max with the RoomPlan app, each containing hundreds of RGB frames. These scenes were also used in the Object-Centric Material Recovery benchmark. However, this benchmark evaluates a fully automated, end-to-end reconstruction pipeline—from raw input scans to a final graphics-ready scene. Lighting is not explicitly estimated; instead, we apply a general environmental light to provide consistent global illumination.

To measure visual quality, we adopt a perceptual similarity metric. Rendered images from the reconstructed scene—viewed under the original camera poses—are compared to real RGB captures.

We evaluate four reconstruction pipelines:

1. Phone2Proc [11] – a procedural reconstruction pipeline for embodied AI, operating at the room level with less emphasis on fidelity;

2. Digital Cousin [9] – integrated with our procedural reconstruction pipeline, but using the Digital Cousin flow for retrieval;

3. Digital Cousin + Make-It-Real (MIR) – enhanced with language-based material selection;

4. Digital Cousin + Ours (Vision+Semantic) – our proposed material selection method using combined vision-language guidance.

# D  More on Methodologies

## D.1  Scene Parsing Stage

In this section, we describe the scene-parsing stage, which utilizes a scene graph to structure spatial relationships. The input includes detected wall structures and oriented object bounding boxes.

This is a rule-based approach, implemented in five key steps:

- Step 1: **Wall Closure**. We start by taking the raw 2D wall segments and snapping any nearly matching endpoints together using a KD-Tree with a small distance threshold to close tiny gaps. We then build a simple graph in which each snapped point is a node and each wall segment is an edge. We traverse this graph to find a closed loop; if no loop exists, we connect the two loose ends and try again. The final loop becomes our room-boundary polygon.

- Step 2: **Wall–Object Alignment**. Next, we examine each object's 2D corners and orientation to determine which wall segment it belongs to. We select the closest wall (within 0.2 m) whose orientation is within 10° of the object's facing direction. We then "snap" the object by moving its anchor point to the nearest point on that wall (leaving a small clearance) and rotating it to align its front face with the wall.

- Step 3: **In-Room Adjustment**. To ensure no object sits outside the room, we test every object corner against the boundary polygon. Any object that falls outside triggers a local expansion of the offending wall segments and reconnection into a single loop. We repeat this until all objects are safely inside.

- Step 4: **Scene Graph Construction.** At this stage, we compute wall connectivity and wall–object relationships. We then identify object–object relationships: if two objects' projections onto the XY plane overlap and their Z extents indicate a clear top–bottom arrangement, we assign an "on-top" link; if two objects share the same orientation and lie within 0.1 m horizontally, we assign a "next-to" link.

- Step 5: **Collision Resolution.** Finally, we resolve any remaining overlaps by treating each object footprint as a 2D polygon. Wall-attached objects are constrained to slide along their wall; others move freely. We iterate up to ten times, find every pair of overlapping objects, compute the minimal translation vector to push them apart (projecting along the wall if constrained), and move them until no collisions remain, or reach the maximum iteration.

We demensarate the results of scene parsing quantitiviely in Figure 17

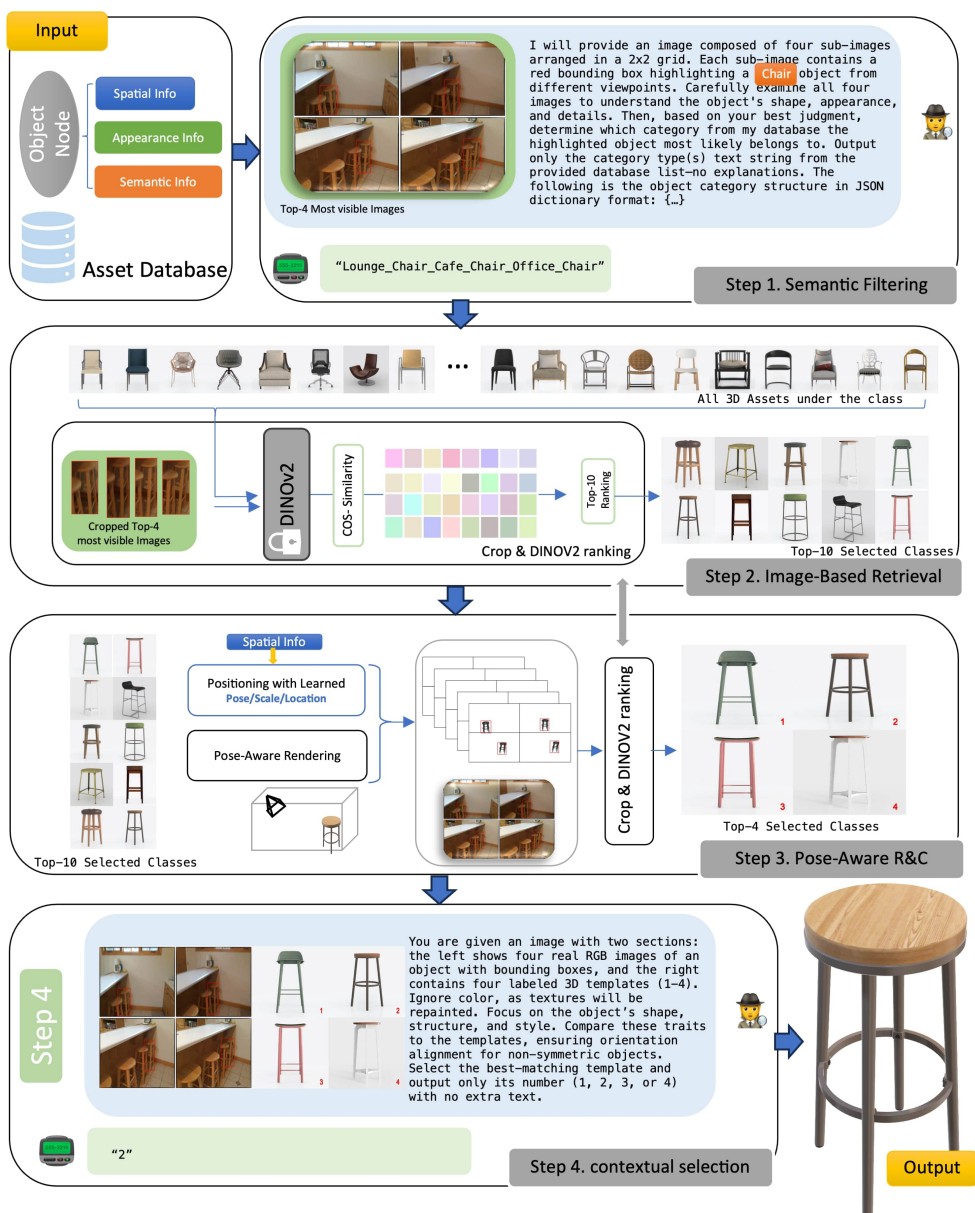

Figure 15: **Detailed Breakdown of Procedural Generation** The input data includes object nodes with spatial, aesthetic, and semantic information, which are utilized at different stages of the pipeline.

## D.2 Retrieval Stage

A detailed breakdown of each step in the proposed retrieval approach is illustrated in Figure 15.

# E   More Results

In this section, we present extended results of LiteReality, focusing particularly on the quantitative evaluation for some scenes in ScanNet, as shown in Figure 10.

For the individual stages of our pipeline, we first showcase the scene parsing results in Figure 17.

Regarding object retrieval, we provide a detailed analysis that includes both the reference images and the corresponding retrieved objects, as illustrated in Figure 18. We also highlight the effectiveness of our contextual selection module, and include some failure cases to demonstrate its limitations.

For the material painting component, we present further comparisons between our proposed method and several baseline models, with results visualized in Figure 16. Also the detailed per categories results tested on the proposed benchmark in Table 5.

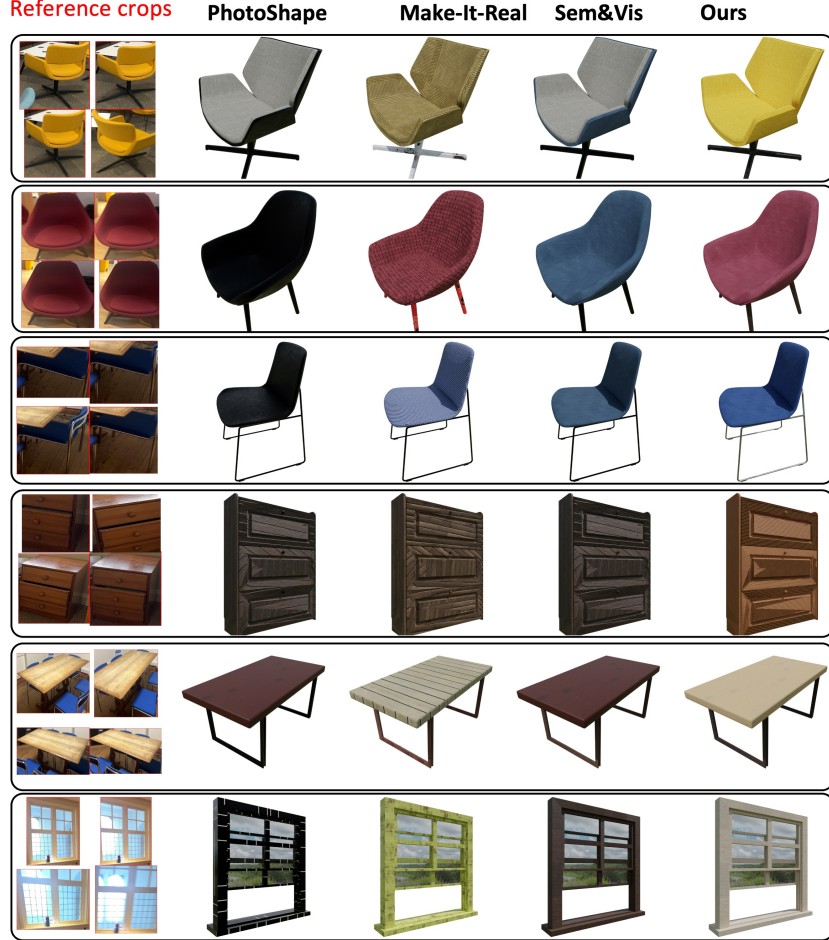

Figure 16: Object-centric material painting results. Our method is compared against several baseline models, demonstrating improved texture alignment and realism.

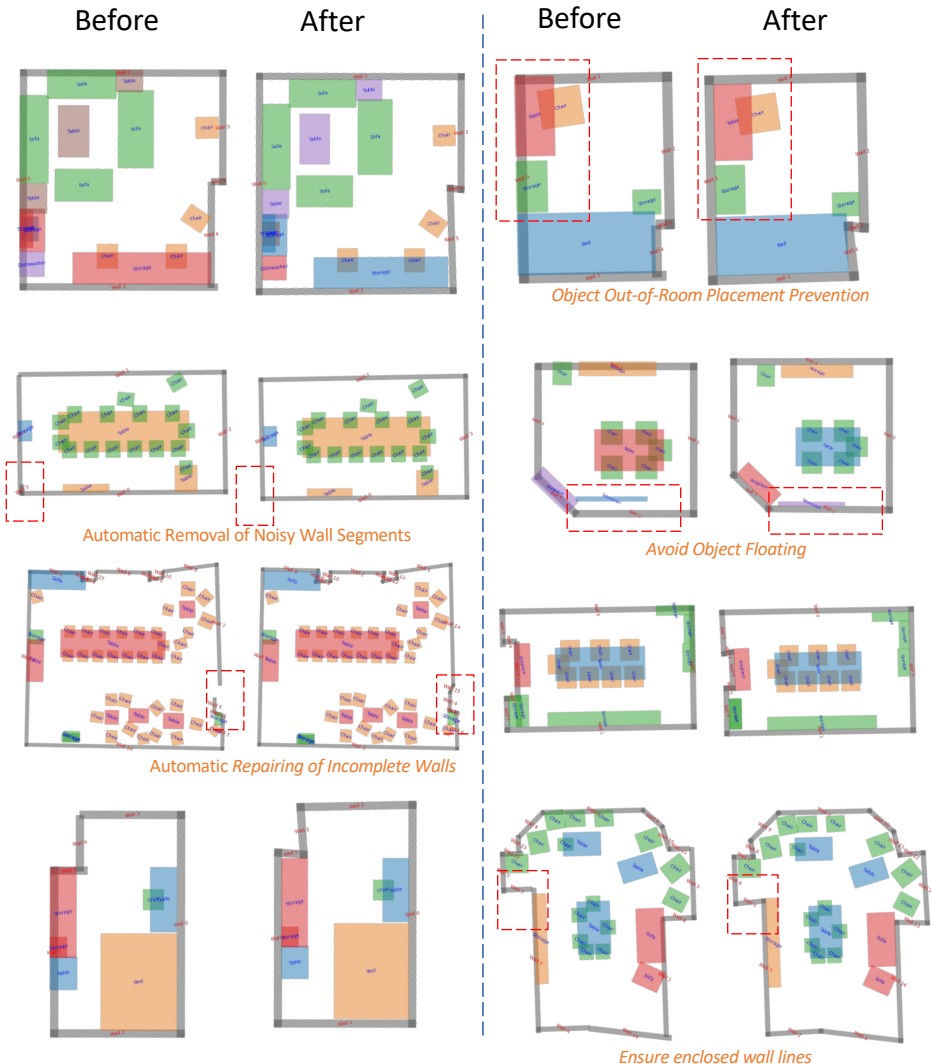

Figure 17: **Scene Parsing Enhancements in LiteReality.** Visualization of several improvements introduced by our scene parsing module. Left and right panels show "Before" and "After" comparisons.

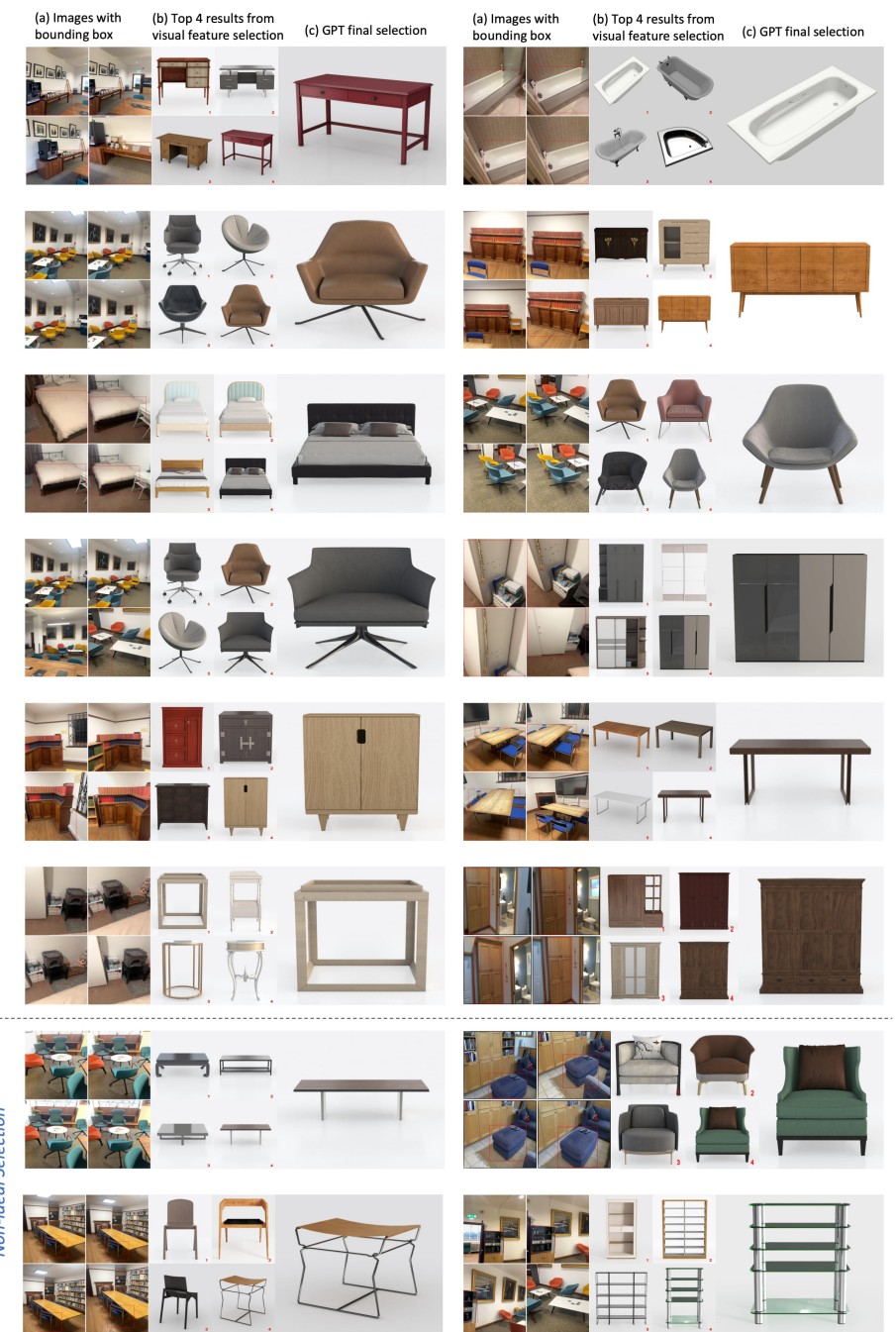

Figure 18: **Demonstration of retrieval results using the proposed multi-step object retrieval pipeline.** The image on the left displays the top four visible images from the dataset. The middle images show the top four ranked results from the visual selection process after Step 3 or Step 2. The right images present the results proposed by GPT-4o. GPT-4o leverages more contextual information, often making more reasonable selections compared to purely visual-based methods. However, failure cases can still occur, as shown in the bottom image, where unreasonable 3D model selection hinders the realism of LiteReality. With advancements in reasoning within MLLMs, we anticipate further improvements in such pipelines in performance as MLLMs continue to evolve.

(a) Chair, Fireplace, Sofa, and Storage

| Method | Chair | | | Fireplace | | | Sofa | | | Storage | | |
|---|---|---|---|---|---|---|---|---|---|---|---|---|
| | RMSE | SSIM | LPIPS | RMSE | SSIM | LPIPS | RMSE | SSIM | LPIPS | RMSE | SSIM | LPIPS |
| PhotoShape | 0.309 | 0.255 | 0.644 | 0.351 | 0.171 | 0.731 | 0.360 | 0.404 | 0.704 | 0.334 | 0.176 | 0.712 |
| MIR+AO | 0.227 | 0.403 | 0.582 | 0.285 | 0.403 | 0.677 | 0.270 | 0.489 | 0.639 | 0.218 | 0.389 | 0.609 |
| MIR | 0.242 | 0.384 | 0.604 | 0.291 | 0.394 | 0.668 | 0.263 | 0.488 | 0.645 | 0.238 | 0.365 | 0.629 |
| Sem&Vis | 0.263 | 0.372 | 0.627 | 0.299 | 0.449 | 0.665 | 0.266 | 0.475 | 0.685 | 0.287 | 0.339 | 0.652 |
| **LiteReality** | **0.228** | **0.407** | **0.578** | **0.283** | **0.433** | **0.688** | **0.270** | **0.486** | **0.634** | **0.230** | **0.395** | **0.621** |

(b) Table, Television, Door, and Windows

| Method | Table | | | Television | | | Door | | | Windows | | |
|---|---|---|---|---|---|---|---|---|---|---|---|---|
| | RMSE | SSIM | LPIPS | RMSE | SSIM | LPIPS | RMSE | SSIM | LPIPS | RMSE | SSIM | LPIPS |
| PhotoShape | 0.401 | 0.304 | 0.640 | 0.351 | 0.242 | 0.652 | 0.259 | 0.311 | 0.658 | 0.441 | 0.215 | 0.750 |
| MIR+AO | 0.250 | 0.467 | 0.576 | 0.219 | 0.639 | 0.537 | 0.214 | 0.507 | 0.590 | 0.289 | 0.323 | 0.692 |
| MIR | 0.256 | 0.448 | 0.600 | 0.218 | 0.632 | 0.543 | 0.207 | 0.503 | 0.603 | 0.332 | 0.285 | 0.689 |
| Sem&Vis | 0.344 | 0.391 | 0.626 | 0.261 | 0.613 | 0.618 | 0.234 | 0.519 | 0.619 | 0.482 | 0.193 | 0.747 |
| **LiteReality** | **0.248** | **0.477** | **0.573** | **0.220** | **0.636** | **0.531** | **0.212** | **0.546** | **0.581** | **0.286** | **0.361** | **0.689** |

Table 5: Per-category evaluation metrics (RMSE ↓, SSIM ↑, LPIPS ↓) for different methods across representative object types. Best results are in bold.

# F    Application of LiteReality

In this section, we showcase how scenes reconstructed with LiteReatliy can be used for a wide range of application, benifits from its graphic realted featuers. The Demensration includes:

**Relighting:** Reconstructed scenes with physically based rendering (PBR) materials support dynamic relighting under various conditions, as demonstrated in Figure 19. This adaptability enhances rendering fidelity and enables realistic, immersive experiences.

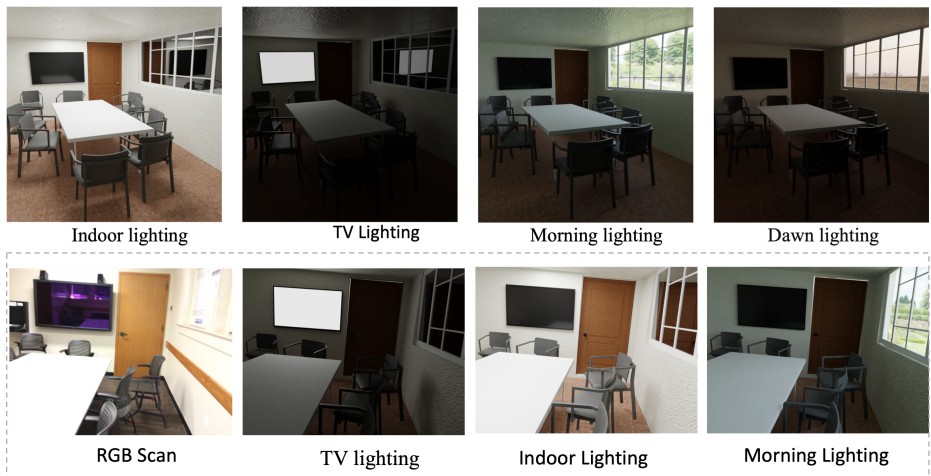

Figure 19: Lighting Effects on Procedural Reconstructions: This figure illustrates the versatility of procedural reconstructions enhanced with PBR materials under various lighting conditions. The top row showcases the reconstructed scene rendered under different lighting setups: standard indoor lighting, TV lighting where the television acts as the sole light source, morning lighting with natural daylight streaming through the window, and dawn lighting that mimics the soft glow of early sunrise. The bottom row provides a comparison to the original RGBD scans used as inputs. This comparison demonstrates the effectiveness of the pipeline in achieving a realistic scene appearance. The ability to seamlessly re-light scenes post-reconstruction highlights the utility of this approach in diverse applications, such as virtual reality, architectural visualization, and cinematic production, where dynamic lighting enhances realism and usability. [Scene is reconstructed from ScanNet scene0166]

**Physics-Based Interactions:** The pipeline enables realistic rigid-body physics and collision detection, allowing natural interactions such as objects falling or colliding. Figure 20 illustrates these capabilities with examples like a falling apple and dynamic indoor collisions.

**Scene Editing:** Modular reconstruction allows users to dynamically replace, modify, or add objects. Integrated physics ensures realistic placement and interaction, simplifying tasks such as adding messy, lifelike elements to a scene (Figure 21).

**Applications in Robotics, Gaming, and VR:** The structured and interactive nature of reconstructed scenes facilitates seamless integration into robotics, gaming, and VR environments. Figure 21 showcases examples that demonstrate the pipeline's potential for creating realistic digital replicas of real-world spaces.

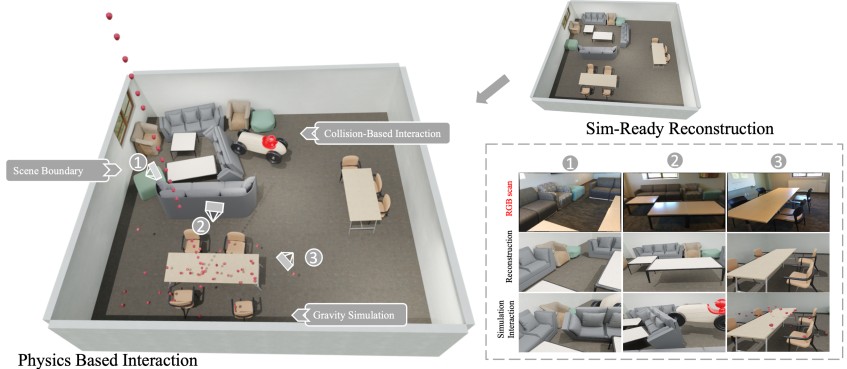

Figure 20: Physics-Based Interaction in Procedural Reconstructions: This figure demonstrates the application of physics simulations to procedurally reconstructed scenes, enabling realistic interactions between objects. The left panel highlights two examples: (1) a gravity simulation where numerous apples are dropped into the scene, naturally landing on tables, scattering, and interacting with the environment; and (2) a toy car driving at high speed, colliding with furniture, showcasing rigid body dynamics as individual objects respond realistically to impacts and forces. The right panel provides a comparison of the original RGB images, reconstructed geometry, and final simulation-ready scenes, emphasizing how the reconstructed objects exhibit realistic physical behaviour. This integration of physics-based interactions enhances the realism and utility of reconstructed scenes, making them suitable for applications in simulation environments, gaming, and robotics.[Scene is reconstructed from ScanNet scene0291]

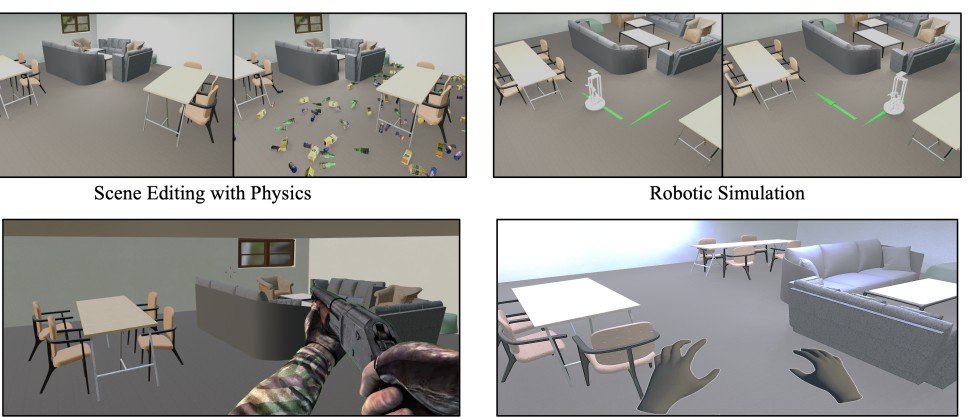

Figure 21: Demonstration of applications enabled by graphics-ready reconstructed scenes, showcasing advanced features such as physics-based interactions, robotic training, gaming environments, and immersive virtual reality experiences. These examples highlight the versatility and interactivity of the reconstructed scenes for various use cases. (Scene is reconstructed from ScanNet *Scene0281*)

