# OpenReview forum: "LiteReality: Graphics-Ready 3D Scene Reconstruction from RGB-D Scans"
_NeurIPS.cc/2025/Conference — NeurIPS 2025 poster_

### Official Review · Reviewer_5sYV · 2025-06-30

**Clarity:** 2
**Significance:** 3
**Originality:** 1
**Rating:** 4
**Confidence:** 4

**Summary:**

This paper proposes a framework that reconstructs digital scenes from RGB-D scans at the “graphics-ready” level. The “graphics-ready” scene provides control of each object, such as furniture and building components. To this end, it proposes a reconstruction pipeline that constructs digital components from the outside in, utilizing prebuilt modules. The fidelity of each object has been enhanced by estimating its material parameters from real photos. The final result is evaluated for three criteria: retrieval similarity, material estimation, and full scene reconstruction. The image metric and geometric difference metrics are used for these criteria.

**Questions:**

I have the following questions:

- What types of metrics should be used to evaluate the proposed method's performance adequately? Does the proposed benchmark thoroughly measure the perceptual (meaningful) performance of methods?

- Are the proposed criteria commonly agreed upon by designers or related industry workers?

- Is each step evaluated individually, and does this paper conduct an ablation study for processing steps?

- Is measuring photorealism for a method that doesn't estimate lighting adequate?

**Ethical Concerns:**

["NO or VERY MINOR ethics concerns only"]

**Final Justification:**

After discussing with the authors and reading other reviewers’ discussions, most of my concerns have been resolved, so I am leaning toward accepting this paper. I appreciate the authors’ thorough and logical replies. This work will demonstrate potential for other downstream tasks.

**Limitations:**

Yes.

**Paper Formatting Concerns:**

In Figure 6, sever -> severe

**Quality:**

2

**Strengths And Weaknesses:**

This paper has the strength that it explicitly introduces and addresses the ultimate goal of the digital twin. The proposed method looks solid and produces a plausible digitization result.

However, there are the following concerns that make me hesitate to support this paper;

First, the topic addressed in this paper is too broad to be covered within the confines of a conference paper. Each step in the pipeline could become the topic of a single paper. This paper should be submitted to graphics journals or HCI journals for completion of this study.

Although the introduction excites me, the method part yields a weak technical novelty, as a multi-modal large language model processes most steps.

The construction of this paper evokes the following question: “Does this paper provide the scientific perspective for the pursued direction?” In this perspective, the proposed evaluation could be significantly improved and theoretically grounded. The current method naively measures the image difference or geometric difference for final evaluation, but these metrics don’t tell us how users perceive the usability of the proposed method. Additionally, the current image metric implicitly evaluates the geometric alignment, texture similarity, and geometric similarity simultaneously, making it challenging to identify which part is problematic.

Although the light is not estimated, the proposed method measures the difference between the rendered image and the ground truth. This means light estimation error distorts albedo estimation. This should be clarified in the paper.

---

> ### Author Rebuttal · Authors · 2025-07-30
>
> We thank the reviewer for the detailed and thoughtful review. We appreciate that the reviewer finds the proposed method solid, acknowledges the quality of the reconstruction results, and recognizes our approach as targeting the ambitious goal of building graphics-ready digital twins. We address the concerns and questions below.
>
>
> **Concern 1: First, the topic addressed in this paper is too broad to be covered within the confines of a conference paper. Each step in the pipeline could become the topic of a single paper. This paper should be submitted to graphics journals or HCI journals for completion of this study.**
>
> Our method specifically targets the problem of **reconstructing indoor 3D scenes into clean, compact, and realistic CAD-like representations with high-quality mesh models and PBR materials, while remaining scalable**. This task is central to spatial AI and has broad applications in data generation, robotics, AR/VR, and simulation. Prior works such as Scan2CAD and SceneCAD have tackled similar end-to-end goals, demonstrating the relevance and suitability of this topic for the conference format. Additionally, recent papers with related system-level designs—such as ProcTHOR (NeurIPS 2022 Outstanding Paper) and Architect (NeurIPS 2024)—have also been published at NeurIPS, further supporting the appropriateness of our submission for this venue.
>
> **Concern 2: Although the introduction excites me, the method part yields a weak technical novelty, as a multi-modal large language model processes most steps.**
>
> We appreciate the reviewer’s interest in the introduction and would like to clarify that LiteReality is not simply a composition of LLM-based modules. In fact, several key components of our system introduce non-trivial technical contributions beyond the use of multi-modal LLMs:
> - Scene Parsing introduces a scene graph–guided layout optimization method to automatically generate physically plausible and well-structured room layouts. This step is critical for realism and usability, yet has been largely overlooked in prior work (see Supplementary Figure 8).
> - The hierarchical object retrieval module, which incorporates pose-aware rendering and comparison, is designed to robustly handle real-world noise and variation. It achieves state-of-the-art results on the Scan2CAD retrieval similarity benchmark (Table 1).
> - The albedo-only optimization scheme offers a novel and practical alternative to full lighting estimation for PBR material recovery. This approach improves scalability while remaining robust under varying illumination (see Figures 6 and 7).
>
> To our knowledge, LiteReality is the first system that produces realistic CAD representations with PBR materials from raw RGB-D scans, addressing several challenges such as occlusion and lighting variation in an integrated and scalable way
> The strength of our pipeline lies in how we redesign and invent components to handle real-world noise, occlusions, geometric misalignments, and lighting variations.
>
> **Concern 3: Does this paper provide a scientific perspective for the pursued direction?**
>
> - LiteReality is, to the best of our knowledge, the first method that reliably reconstructs indoor scenes into compact CAD representations with realistic PBR materials and parsed layouts.
> - We also build three rigorous benchmarks to evaluate retrieval similarity, object-centric material recovery, and overall realism, supporting the development of this research direction. [Supp Section C]
> - We provide a curated database containing manually cleaned and polished material segmentation for thousands of indoor furniture items, supporting retrieval and material-painting–based object reconstruction. [Supp Section B]
>
>
> **Concern 4: The proposed evaluation could be improved and theoretically grounded. Current metrics naively measure image and geometric differences and don’t capture user perceptions of usability.**
>
> Our evaluation metrics follow a well-established line of work in CAD retrieval and appearance modeling—we simply adopt the established metrics used in prior work. Methods such as Scan2CAD, DiffCAD, and MSCD all employ same retriveal silmirity metrics. For appearance modeling, perceptual accuracy metrics are widely used in PSDR-Room, PhotoScene, and other works that aim to recover appearance from images. While we acknowledge that no metric is perfect, these have shown strong correlation with human perception in prior literature and are commonly adopted in related works.
>
>
> **Concerns 5: Additionally, the current image metric implicitly evaluates the geometric alignment, texture similarity, and geometric similarity simultaneously, making it challenging to identify which part is problematic.**
>
> Table 1 and Table 2 in the paper evaluate the performance of two core modules independently in a controlled setting. Table 1 focuses solely on geometry similarity. In Table 2, we use the same retrieved geometry, with the only difference being the material painting methods, thereby isolating the performance of material painting. Table 3, on the other hand, is intentionally designed to jointly evaluate multiple aspects—geometry, texture, and alignment—to reflect the holistic performance of the entire system.
>
> **Concern 6: Although the light is not estimated, the proposed method measures the difference between the rendered image and the ground truth. This means light estimation error distorts albedo estimation. This should be clarified in the paper.**
>
> [Line 300 in main paper], we add lighting through global illumination using an HDR environment. Our PBR material estimation is performed without explicit lighting estimation, as experiments with lighting components were found to be less scalable and introduced significant computational overhead. Theoretically, this omission may introduce some distortion in albedo estimation. Our material selection module leverages both visual cues and language descriptions, and demonstrates encouraging results even without explicit lighting estimation (see Figures 6 and 7). We will clarify this point in the revision.
>
> **Question 1: What types of metrics should be used to evaluate the proposed method's performance adequately? Does the proposed benchmark thoroughly measure the perceptual (meaningful) performance of methods? Are the proposed criteria commonly agreed upon by designers or related industry workers?**
>
> - These metrics used in the paper are widely adopted and recognized in the community for evaluating both retrieval quality (DiffCAD, ScanNotate,  HOC-Search) and appearance modeling (PSDR-Room, Photo-Scene).
> - Additionally, we conducted a user study to evaluate both the retrieval and material-recovery components of our system. six  participants—graphics researcher, graphics engineer, architect, and 3D-vision specialists—each viewed four real-world scenes and were allowed to select up to two favorite results from all methods under comparison. Tables below show the percentage of favorite votes for each method in both tasks. Our proposed approach achieved the highest vote share in both retrieval and material painting, in line with the evaluation matrix presented earlier.
>
>
>  | Scene                   | LiteReality        | DC                 | Phone2Proc          |
> |-------------------------|-------------------:|-------------------:|--------------------:|
> | **Boardroom**           | 31 (46.3 %)        | 30 (44.8 %)        | 6 (9.0 %)           |
> | **Common Room**  | 10 (47.6 %)        | 7 (33.3 %)         | 4 (19.0 %)          |
> | **Study Room**            | 7 (46.7 %)         | 7 (46.7 %)         | 1 (6.7 %)           |
> | **Meeting Room**                 | 13 (61.9 %)        | 6 (28.6 %)         | 2 (9.5 %)           |
> | **Total**               | 61 (49.2 %)    | 50 (40.3 %)    | **13 (10.5 %)**     |
>
>
> | Scene                   | LiteReality | MIR + AO | Sem&Vis | MIR   | PhotoShape |
> |-------------------------|----------------------:|----------:|----------------:|------:|-----------:|
> | **Boardroom**           | **81.4 %**                | 62.4 %    | 2.7 %           | 5.3 % | 7.2 %      |
> | **Common Room**  | 45.3 %                | **61.8 %**    | 15.3 %          | 1.8 % | 3.4 %      |
> | **Meeting Room**          | **57.8 %**                | 33.9 %    | 1.9 %           | 3.6 % | 4.2 %      |
> | **Study Room**    | **57.2 %**                | 15.9 %    | 3.0 %           | 20.4 %| 5.1 %      |
>
> **Question 2: Is each step evaluated individually, and does this paper conduct an ablation study for processing steps?**
>
> Table 1 and Table 2 evaluate the two core modules individually, while Table 3 reflects the overall system performance.
>
> **Question 3: Is measuring photorealism for a method that doesn't estimate lighting adequate?**
>
> Omitting lighting estimation is currently a practical and effective solution for PBR material estimation, as it allows the system to remain robust and scalable. However, we fully agree with the reviewer that incorporating lighting estimation would be a meaningful and valuable extension of this work, and would enhance the realism of the estimated PBR materials. For example, one could train a lightweight neural network to estimate HDR environment maps or point light positions directly from RGB-D inputs, which can then be used to refine material appearance. We will include a thorough discussion of this direction in the paper.
>
> Thank you again for your time and thoughtful review of our work.

---

> > ### Comment · Reviewer_5sYV · 2025-08-06
> >
> > Thank the authors for their rebuttal. Here I attach my comments.
> >
> > Concern 1: First, the topic addressed in this paper is too broad to be covered within the confines of a conference paper. Each step in the pipeline could become the topic of a single paper. This paper should be submitted to graphics journals or HCI journals for completion of this study.
> >
> > Comments: Thank you for the research context.
> >
> > Concern 2: Although the introduction excites me, the method part yields a weak technical novelty, as a multi-modal large language model processes most steps.
> >
> > Comments: I understand.
> >
> > Concern 3: Does this paper provide a scientific perspective for the pursued direction?
> >
> > Comment: I agree that building a whole automation pipeline is novel.
> >
> > Concern 4: The proposed evaluation could be improved and theoretically grounded. Current metrics naively measure image and geometric differences and don’t capture user perceptions of usability.
> >
> > Comment: Measuring the color similarity of images of whole objects as material estimation accuracy has a high correlation but also an inevitable bias due to errors in early steps. How about measuring patch similarity on the corresponding surfaces, or at least the difference at corresponding points? Additionally, it would be nice to use a vision-language model to measure the semantic similarity between the reconstructions and the observations.
> > Also, I would like to know how it handles similarity for spatially-varying BRDFs, such as the one shown on the sofa in the teaser.
> >
> > Concerns 5: Additionally, the current image metric implicitly evaluates the geometric alignment, texture similarity, and geometric similarity simultaneously, making it challenging to identify which part is problematic.
> >
> > Comment: Thanks for the clarification. Comparison with the same retrieved geometry looks sound.
> >
> > Question 1: What types of metrics should be used to evaluate the proposed method's performance adequately? Does the proposed benchmark thoroughly measure the perceptual (meaningful) performance of methods? Are the proposed criteria commonly agreed upon by designers or related industry workers?
> >
> > Comment: Thank you for the experiment.
> >
> > Question 3: Is measuring photorealism for a method that doesn't estimate lighting adequate?
> >
> > Concern 6: Although the light is not estimated, the proposed method measures the difference between the rendered image and the ground truth. This means light estimation error distorts albedo estimation. This should be clarified in the paper.
> >
> > Comment: Thank you for including the discussion about lighting.

---

> ### Author Response · Authors · 2025-08-06
>
> We thank the reviewer for the positive feedback on our responses to most of the concerns and questions. Regarding the remaining questions under Concern 4, please find our response below:
>
> **Question 1: How about measuring patch similarity on the corresponding surfaces, or at least the difference at corresponding points for color estimation?**
>
> We have indeed experimented extensively with this approach during the development process. While it is possible, in practice, this method produces suboptimal results due to challenges inherent in real-world scanned data. Specifically:
>
> In the material painting stage, we begin by extracting patch crops from multi-view RGB images for each material segment of the retrieved asset (see Figure 4, part (b)). This is done in two steps: 1. [GroundingDINO + SAM + Filtering] is used to identify candidate patch regions. 2. MLLM then selects the most representative patches using chain-of-thought (CoT) prompts. [This method has proven to be more robust than prior patch selection approaches used in material estimation (e.g., in PSDR-Room and PhotoScene), particularly when applied to large-scale scans.] In other words, patch-level information exists for each segmentation and can be used for color querying.
>
> **However, in practice, directly measuring color from the patches—even with multiple reference patches—is not ideal, as lighting is inherently entangled with material appearance.** Many patches appear overly gray or dark due to poor lighting conditions, and using these for color estimation often results in unrealistic, desaturated outputs—sometimes even pure gray.
>
> We perform color querying by inputting both the full image and the highlighted material segment into the MLLM, enabling global reasoning based on the reference image. This process is repeated N times, as each object is associated with N crops from different reference images, to increase robustness and proved to work well.
>
> **Question 2, Additionally, it would be nice to use a vision-language model to measure the semantic similarity between the reconstructions and the observations.**
>
> We agree this is a meaningful metric, as it captures semantic similarity between scenes. It is similar to the background consistency metric in VBench, which uses CLIP features to evaluate frame-level consistency [CVPR 2024 Highlight].
>
> To explore this idea, we conducted a preliminary experiment measuring the semantic similarity between the reconstructed scenes and the corresponding RGB images inputs. We used CLIP features for all frames rendered with the same camera poses (as shown in the vertically split-screen demos in the video). Due to time constraints, we only tested this on two scenes and compared the results with other baseline methods. Nevertheless, the results were promising: LiteReality achieved the highest cosine similarity in CLIP visual features. A more comprehensive study will be included in the final revision of the paper.
>
> | Scene      | Phone2Proc | Digital Cousin | DC + MIR | **LiteReality** |
> |-|---|--|----------|-|
> | Study Room | 0.7144     | 0.7099         | 0.6879   | **0.7226**      |
> | Bedroom    | 0.6728     | 0.6730         | 0.6653   | **0.6842**      |
>
> Both scenes appear at 3:10 in the demo video. This metric also reflects reconstruction completeness—more detailed scenes reconstruction with small and wall-mounted objects are likely to score higher. Thanks for the suggestion.
>
> **Question 3: Also, I would like to know how it handles similarity for spatially-varying BRDFs, such as the one shown on the sofa in the teaser.**
>
> Please correct me if I’ve misunderstood your question. the question is how the system handles similarity for spatially-varying BRDFs (SVBRDFs) during material selection, i.e. how it selects the most appropriate BRDF pattern for a given material segment.
>
> Two prior works are particularly relevant: 1. MatSynth (CVPR 2024), (a large-scale PBR material database) 2.Make It Real (NeurIPS 2024), (language descriptions for each material in MatSynth
>
> For each segmented material region with its reference images, we first perform robust multi-view patch cropping (as described above). We then conduct a language-based retrieval using a three-step querying process in the PBR database:
> [Overall material type → More specific subtype → Top 10 candidate materials]  This typically selects the correct material type, although the visual appearance may still differ.
>
> Then, we compare the 10 rendered PBR candidates with the cropped patches using CLIP, select the top 4, and let an MLLM pick the final match using contextual cues.
>
> This pipeline—progressing from material type prediction to visual similarity filtering and MLLM-based refinement—ensures a reliable initialization from MatSynth. The final albedo-only optimization adjusts the appearance to match the real-world reference while preserving the high-quality BRDF from the retrieved material.
>
> Thanks again and do let us know if you have further questions.

---

> ### Author Response · Authors · 2025-08-08
>
> Dear Reviewer,
>
> Thank you again for your thoughtful review and for engaging with our rebuttal. Since the discussion period is nearly over, we would like to check if your remaining questions have been addressed in our last response. If there is anything else you would like clarified, please feel free to let us know.
>
> Lastly, since most concerns — including yours and those from other reviewers (**jpwk**, **meLw**， **y37B**) — have now been addressed, we would be grateful if you might consider updating your score if appropriate.
>
> Best regards,
> The LiteReality Team

---

### Official Review · Reviewer_y37B · 2025-07-01

**Clarity:** 3
**Significance:** 3
**Originality:** 2
**Rating:** 4
**Confidence:** 4

**Summary:**

This paper presents a pipeline for transforming RGB-D scans of indoor environments into detailed 3D virtual replicas. The approach begins with scene understanding, involving object detection and 3D bounding box prediction, followed by the construction of a scene graph to capture spatial relationships between scene elements. For each detected object, a corresponding artist-crafted 3D model is retrieved from a 3D asset database using a Multi-modal Large Language Model (MLLM) that leverages both visual and textual cues. The method then estimates materials and textures for each object using MLLM-based techniques, enabling realistic material painting. Finally, the entire scene is reconstructed in Blender for photorealistic rendering. The pipeline is evaluated across three tasks: object retrieval accuracy, material estimation quality, and procedural generation of the complete scene. This method outperforms the state-of-the-art methods in all three tasks.

**Questions:**

1. Dependency on Camera Pose Estimation:
It is unclear which camera pose estimation model is employed in the pipeline. Since accurate object placement relies heavily on correct pose information, it would be helpful to discuss how object retrieval behaves when the pose estimation is inaccurate. An ablation or sensitivity analysis demonstrating the effect of pose estimation errors on object retrieval accuracy would strengthen the paper.

2. Scene Completeness:
The final reconstructed scenes appear to be missing several objects, which affects the perceived completeness and realism. A quantitative evaluation of scene completeness—such as the percentage of detected objects successfully reconstructed—would provide valuable insight into the robustness and practical usability of the method.

3. Evaluation Metrics and User Study:
While Chamfer Distance and image-based metrics like LPIPS and PSNR are informative, they may not fully capture perceptual quality or semantic accuracy. Including a user preference study would provide complementary qualitative insights, especially for evaluating object retrieval and material estimation results.

**Ethical Concerns:**

["NO or VERY MINOR ethics concerns only"]

**Final Justification:**

Although the paper’s pipeline offers limited technical novelty, the problem setup—producing realistic CAD representations with PBR materials from raw RGB-D scans at the room level—is itself an important and non-trivial research direction. Conceiving such a setup from individual existing components is not straightforward, and therefore, I am inclined to recommend acceptance of the paper.

**Limitations:**

Discussed in the supplementary material.

**Paper Formatting Concerns:**

No major paper formatting concern.

**Quality:**

3

**Strengths And Weaknesses:**

Strengths:

1. The paper addresses an important research problem—generating virtual 3D replicas of indoor scenes—which has broad applications in areas such as AR/VR, robotics, and simulation.

2. The paper is clearly written, and the detailed explanation of the proposed pipeline enhances reproducibility and facilitates future research.

Weaknesses:

 1. Limited Novelty: The proposed method primarily integrates existing components—object retrieval and object-centric material estimation—into a unified pipeline. While the combination is useful, the individual components are largely based on prior work, and the paper offers limited methodological innovation.

 2. Insufficient Evaluation:
a) The paper lacks a comprehensive evaluation of the reconstructed scenes. For instance, in the object retrieval task, it does not analyze discrepancies in object scale or orientation compared to ground-truth assets (e.g., in Figure 5, second row, last example, the retrieved table differs noticeably in size).
b) It is unclear whether the objects are aligned with ground-truth counterparts before computing Chamfer Distance. If alignment is performed, the metric may fail to reflect orientation mismatches, which are important for realistic reconstruction.

---

> ### Author Rebuttal · Authors · 2025-07-31
>
> We thank the reviewer for the positive and constructive feedback. We are glad that the reviewer found our work technically solid and impactful for applications in AR/VR, robotics, and simulation. Below, we respond to the specific concerns raised:
>
> **W1: Limited methodological innovation—primarily an integration of existing modules**
>
> The primary goal of this work is to **enable compact, graphics-ready 3D reconstruction from real-world scans at the room level—featuring high-quality geometry and realistic PBR materials**.  It is important to highlight that simply combining existing modules does not lead to usable results, let alone robust ones. The strength of our pipeline lies in how we redesign and invent new components to handle real-world noise, occlusions, geometric misalignments, and lighting variations. Specifically,
> - We introduce a scalable, robust, and self-contained PBR material recovery module that performs reliably even under poor lighting conditions, severe geometric misalignments, and heavy occlusions—challenges commonly encountered in real-world scans. Our method achieves state-of-the-art fidelity with a significant margin over baseline methods (Table 2 and Table 3).
> - We also achieve state-of-the-art retrieval similarity performance, rigorously benchmarked on the Scan2CAD dataset (Table 1), demonstrating substantial improvement in average Chamfer Distance per CAD over previous methods and achieving the best performance across nearly all object categories.
>
>
>
> **W2 a) Lack of object scale/orientation analysis in retrieval**
>
> In the pipeline, the retrieval stage was conducted with off-the-shelf methods (here with Apple RoomPlan) to detect oriented bounding boxes. According to their report, RoomPlan offers 91% average precision / 90% recall at 3D IoU 30% for object detection. The error was introduced there. Since it is not part of our contribution, evaluation metrics are not proposed.That said, we agree that analyzing alignment and scale discrepancies (e.g., as done in Scan2CAD) would enhance the evaluation, and we plan to incorporate such analyses in future work.
>
> **W2 b) Alignment in Chamfer Distance evaluation**
> We do not perform any alignment before computing Chamfer Distance. All 3D assets in our database are calibrated to have a canonical orientation (facing forward) and are positioned in the scene using the orientation provided by the 3D oriented Bbox.
>
> **Q1: Dependency on Camera Pose Estimation**
>
> Camera pose is obtained from off-the-shelf models used in the scene understanding stage. Pose information is primarily used at two points during the retrieval
> 1.To project 3D bounding boxes into 2D images for object image cropping.
> 2. In the Pose R&C step, to position objects within the scene and generate renderings for more effective candidate filtering.
>
> - For point 1, our implementation includes an optimization step that uses GroundingDINO to refine the back-projected 2D bounding boxes by aligning them with overlapping detection results from GroundingDINO. This yields a highly robust cropping scheme. As a result, the effect of camera pose errors for this is significantly mitigated.
> - For point 2, we showcase ablation results below comparing retrieval similarity with and without the Pose R&C stage. A general drop in retrieval similarity is observed, but the system still performs well. This ablation is conduct with similar manner as Table 1.
>
> | ID  | Semantic | Image Feat. | Pose R&C | MLLM | Avg. Chamfer (↓) |
> |-----|----------|-------------|----------|------|------------------|
> | I   | ✓        | ✓           |          |      | 0.142            |
> | II  | ✓        | ✓           | ✓        |      | 0.110            |
> | III | ✓        | ✓           |          | ✓    | 0.112            |
> | IV  | ✓        |             | ✓        |      | 0.104            |
> | V   | ✓        | ✓           | ✓        | ✓    | **0.097**        |
>
> **Q2: Scene Completeness Evaluation**
>
> The completeness of the reconstructed scene is limited by the capabilities of the scene understanding model. Our system uses Apple RoomPlan, which currently supports 17 object categories (i.e. Bed, Sofa, Chair, Table, Desk, TV, TV Stand, Sink, Bathtub, Toilet, Refrigerator, Stove, Kitchen Counter, Cabinet, Dresser, Nightstand, Bookshelf, Wardrobe, and Fireplace). However, it does not support small objects or wall-mounted items such as lamps or picture frames, which introduces incompleteness in the final scene.
>
> For the supported categories, below is a Detection Quality Table evaluated across four real-life scans. Most objects are accurately detected, with one notable false positive—where a seating area near a window was misclassified as a sofa—and one false negative involving a missed chair in a meeting room.
>
> In the future, we plan to extend object coverage to include smaller and wall-mounted items. We also recognize the value of a completeness metric for evaluation, which can be supported by providing manual annotations of the objects in real world scans
>
> | Scene | Total | TP | FP | FN |
> |-|-|-|-|-|
> | Meeting Room | 21    | 21 | 0  | 1  |
> | Boardroom    | 48    | 48 | 1  | 0  |
> | Common Room  | 20    | 20 | 0  | 0  |
> | Study Room   | 12    | 12 | 0  | 0  |
>
> **Q 3 : Evaluation Metrics and User Study**
>
> To evaluate both retrieval and material estimation, we conducted a small-scale user study on four real-world scenes reconstructed entirely without human intervention. Due to time constraints, the current user study is limited in scale. We plan to conduct a more comprehensive user study after the rebuttal phase.
>
> Study Setup:
> Six Participants (graphics researchers, engineers, architects, and 3D vision researchers) were shown:
> - A reference image of a real-world object.
> - Outputs from different methods (randomized order), including:
> - Retrieval: LiteReality, Phone2Proc, and Digital Cousin.
> - Material: All methods in Table 2.
> Participants selected up to two preferred results per object. The results show that LiteReality was favored in the user study for both the retrieval and material painting stages,
>
> For Retrieval
>  | Scene                   | LiteReality        | DC                 | Phone2Proc          |
> |-------------------------|-------------------:|-------------------:|--------------------:|
> | **Boardroom**           | 31 (46.3 %)        | 30 (44.8 %)        | 6 (9.0 %)           |
> | **Common Room**  | 10 (47.6 %)        | 7 (33.3 %)         | 4 (19.0 %)          |
> | **Study Room**            | 7 (46.7 %)         | 7 (46.7 %)         | 1 (6.7 %)           |
> | **Meeting Room**                 | 13 (61.9 %)        | 6 (28.6 %)         | 2 (9.5 %)           |
> | **Total**               | **61 (49.2 %)**    | 50 (40.3 %)    | 13 (10.5 %)     |
>
> For Materail Painting
> | Scene                   | LiteReality | MIR + AO | Sem&Vis | MIR   | PhotoShape |
> |-------------------------|----------------------:|----------:|----------------:|------:|-----------:|
> | **Boardroom**           | **81.4 %**                | 62.4 %    | 2.7 %           | 5.3 % | 7.2 %      |
> | **Common Room**  | 45.3 %                | **61.8 %**    | 15.3 %          | 1.8 % | 3.4 %      |
> | **Meeting Room**          | **57.8 %**                | 33.9 %    | 1.9 %           | 3.6 % | 4.2 %      |
> | **Study Room**    | **57.2 %**                | 15.9 %    | 3.0 %           | 20.4 %| 5.1 %      |
>
> Thank you again for the thoughtful review and supporting our work.

---

> > ### Author Response · Authors · 2025-08-07
> >
> > Dear Reviewer,
> >
> > Thank you again for your support of our work. Please feel free to let us know if your questions have been answered or if any further clarification is needed.
> >
> > Best regards,
> > The LiteReality Team

---

> > > ### Comment · Reviewer_y37B · 2025-08-09
> > >
> > > Dear Authors,
> > >
> > > Thank you for the detailed explanations and additional experiments. All of my concerns are addressed. I will keep my initial ratings.

---

### Official Review · Reviewer_meLw · 2025-07-01

**Clarity:** 3
**Significance:** 3
**Originality:** 2
**Rating:** 4
**Confidence:** 4

**Summary:**

This paper presents a framework for reconstructing indoor scenes from RGB-D scans into articulated geometry with realistic materials, enabling both high-quality rendering and physical interactions. The pipeline comprises four stages: scene parsing, object reconstruction, material painting, and procedural reconstruction. Notably, it introduces a training-free object retrieval method and a robust material painting module, both achieving state-of-the-art performance.

**Questions:**

Question:
1.	Is it a fair comparison in Table 3 given that the proposed method uses RGB-D input but the methods like Phone2Proc and Digital Cousin use RGB input?
2.	What is the computation time of each stage, and compare to other baselines?

**Ethical Concerns:**

["NO or VERY MINOR ethics concerns only"]

**Final Justification:**

Most of my concerns have been addressed in the rebuttal. While I still share some reservations about the technical contribution, as noted by other reviewers, the paper shows a degree of originality in relation to prior work and potential applications. Therefore, I have decided to raise my score.

**Limitations:**

yes

**Quality:**

3

**Strengths And Weaknesses:**

Strengths:
1.	The paper is well-written, with a clear presentation of the pipeline and clear visual demonstrations.
2.	The proposed “graphics-ready” 3D reconstruction framework has broad application potential in AR/VR, gaming, and robotics.
3.	Experimental results show consistent improvements over prior works in retrieval accuracy, material recovery, and full-scene realism.

Weaknesses:
1.	Although the authors claim to be the first to convert room-level scans into graphics-ready environments, prior works such as Digital Cousin [1] and Phone2Proc [2] address similar goals and concepts.
2.	Many components of the pipeline are built upon existing methods; the novelty lies primarily in system integration rather than in technical innovation. As such, the contribution is more engineering-driven than algorithmically novel.
3.	The quantitative gains in Tables 2 and 3 are relatively small, and the absolute performance metrics for all methods remain low, raising concerns about the practical impact of the improvements.


[1] Automated creation of digital cousins for robust policy learning. CoRL 2024
[2] Phone2proc: Bringing robust robots into our chaotic world. CVPR 2023

---

> ### Author Rebuttal · Authors · 2025-07-30
>
> We thank the reviewer for the thoughtful and constructive feedback. We appreciate the reviewer highlighting the clarity of our paper, the strong visual and quantitative performance, and the broad applicability of the proposed method. Below, we address each point in detail.
>
> **Weaknesses 1: Although the authors claim to be the first to convert room-level scans into graphics-ready environments, prior works such as Digital Cousin [1] and Phone2Proc [2] address similar goals and concepts.**
>
> we would like to clarify that neither method is designed to produce **realistic CAD representation with PBR materails from room-level scans**:
>
> - **Digital Cousin** works on a **single image** and aims to create simulatable scenes with objects semantically matched to reality. However, it does not support PBR material recovery from images and cannot robustly handle room-level reconstruction from video inputs. [Note that the Table 3 Digital Cousin baseline is built by replacing the retrieval step in our pipeline with theirs, while the rest of the pipeline uses LiteReality.]
> - **Phone2Proc** reconstructs room-level scenes from videos but lacks support for realistic PBR materials, as it is designed primarily for generating diverse simulation environments for robotic training.
>
> - In contrast, our goal is to reliably reconstruct indoor scans into graphics-ready 3D scenes (realistic CAD with PBR materials and parsed layout) at the room level—a task that neither Digital Cousin nor Phone2Proc is able to accomplish (detailed in the table below).
>
> | Method            | Input Type   | Semantic Matching for Retrieval | PBR Materials for Realism | Room-Level Reconstruction |
> |-------------------|--------------|---------------------------------|---------------------------|---------------------------|
> | **Digital Cousin**| single Image    | ✅ Yes                          | ❌ No                     | ❌ No                      |
> | **Phone2Proc**    | RGB-D Video  | ✅ Yes                          | ❌ No                     | ✅ Yes                    |
> | **PSDR-Room**     | single Image    | ✅ Yes                          | ✅ Yes                    | ❌ No                     |
> | **LiteReality**   | RGB-D Video  | ✅ Yes                          | ✅ Yes                    | ✅ Yes                    |
>
> A similar work worth mentioning is PSDR-Room, which shares a similar goal of producing realistic scenes. However, it only works on a single image and cannot be easily scaled to room-level reconstruction.
>
> **Weaknesses 2. Many components of the pipeline are built upon existing methods; the novelty lies primarily in system integration rather than in technical innovation. As such, the contribution is more engineering-driven than algorithmically novel.**
>
> The goal of this work is to enable compact reconstruction from real-world scans at room level with with high quality geomtry mesh and relastic PBR materails—a practically important but previously unsolved problem. This motivation drives the design of LiteReality, which necessarily involves integration of multiple components.
>
> It is important to highlight that simply combining existing modules does not lead to usable results, let alone robust ones. Prior to LiteReality, no method could take raw RGB-D scans and output realistic CAD representations of room-scale 3D scenes with PBR materials. The strength of our pipeline lies in how we redesign and invent components to handle real-world noise, occlusions, geometric misalignments, and lighting variations.
>
> Also, for indivisual compoenent:
> - We introduce a scalable, robust PBR material recovery module that performs reliablely at scale—crucial for realistic room-scale reconstruction—and achieves state-of-the-art fidelity compared to existing methods with large margin [Table2M, Table3M].
> - We also achieve state-of-the-art retrieval performance using a training-free module that integrates semantic filtering and pose reasoning [Table1] with a substantial improvement (around 5%) over previous SOTA, and best performance on nealry very object categores.
>
> We value the delivery of a complete, reliable pipeline more than isolated novelty in individual components, as it enables a new class of applications in AR/VR, data creation, and simulation—which we believe is ultimately highly impactful.
>
> **Weaknesses3. The quantitative gains in Tables 2 and 3 are relatively small, and the absolute performance metrics for all methods remain low, raising concerns about the practical impact of the improvements.**
>
> We would like to highlight that numbers in perceptual quality metric for reconstruction using retrieved geometry and PBR material recovery typically falls substantially short of novel-view synthesis methods like NeRF and Gaussian Splatting. For example, **PSDR-room reports LPIPS ≈ 0.5–0.8 and SSIM ≈ 0.4–0.6—on par with our results—whereas NeRF-based approaches typically achieve LPIPS ≈ 0.1 and SSIM > 0.9**. Nevertheless, this remains an informative metric for demonstrating improvement, as commonly used in prior works. The qualitative results (Figures 5, 6, 7, Supp Figure 1, demo video) demonstrate the clear visual improvements.
>
> Meanwhile, during the rebuttal period, we conducted a user study asking participants to vote for the perferred PBR recovery method. Our proposed method showed a clear advantage in this study.
>
> | Method           | Boardroom       | Common Room     | MeetingRoom      | StudyRoom | Overall      |
> |------------------|----------------:|----------------:|----------------:|-----------------:|-------------:|
> | MIR + AO         | 62.4 %          | **61.8 %**      | 33.9 %          | 15.9 %           | 43.5 %       |
> | Sem & Vis        | 2.7 %           | 15.3 %          | 1.9 %           | 3.0 %            | 5.7 %        |
> | MIR              | 5.3 %           | 1.8 %           | 3.6 %           | 20.4 %           | 7.8 %        |
> | PhotoShape       | 7.2 %           | 3.4 %           | 4.2 %           | 5.1 %            | 5.0 %        |
> | **LiteReality**  | **81.4 %**      | 45.3 %          | **57.8 %**      | **57.2 %**       | **60.4 %**   |
>
> **Q1: 1. Is it a fair comparison in Table 3 given that the proposed method uses RGB-D input but the methods like Phone2Proc and Digital Cousin use RGB input?**
>
> Phone2Proc uses RGB-D videos as input, whereas Digital Cousin only supports RGB images and cannot perform room-level reconstruction. For more details, please see our response to Weaknesses 1.
>
> **Q2. What is the computation time of each stage, and compare to other baselines?**
>
> Our methods runs on a single NVIDIA RTX 3090 GPU. Below is the runtime breakdown per scene:
>
> | Scene        | Preprocessing + Scene Parsing | Object Retrieval | Material Painting | Procedural Reconstruction |
> |--------------|-------------------------------|------------------|-------------------|----------------------------|
> | Boardroom    | 3m30s                          | 4m30s            | 51m               | 1m10s                      |
> | Common Room  | 1m20s                          | 3m39s            | 19m               | 48s                        |
> | Study Room   | 1m50s                          | 2m20s            | 14m               | 45s                        |
> | Meeting Room | 1m20s                          | 3m39s            | 19m               | 48s                        |
>
> Baselines that skip material painting would take less time, but they lack PBR material recovery, a essential part for realistic reconstruction and downstream applications.
>
> Thank you again for the thoughtful review.

---

> > ### Comment · Reviewer_meLw · 2025-08-05
> >
> > Thank you to the authors for their response and the additional experimental results. My concerns regarding Q1 and Q2 have been addressed. After reviewing the other reviewers' comments, I confirm that I am ready to proceed to the reviewer-AC discussion phase.

---

> > > ### Author Response · Authors · 2025-08-09
> > >
> > > Dear Reviewer,
> > >
> > > Thank you for letting us know, and we’re glad to hear that your concerns have been addressed.
> > >
> > > For the reviewer–AC discussion phase, we would like to kindly draw your attention to the recent updates in the discussion with other reviewers since your last reply. In general, the concerns raised have been largely addressed, and all other reviewers have recognized the novelty and meaningfulness of **building an robust automated pipeline to convert RGB-D scans into compact and realistic CAD representations with PBR materials**.
> > >
> > > We will be watching closely in these final hours and are happy to answer any further questions you may have.
> > >
> > > Best,
> > > The LiteReality Team

---

### Official Review · Reviewer_jpwk · 2025-07-03

**Clarity:** 3
**Significance:** 2
**Originality:** 1
**Rating:** 4
**Confidence:** 3

**Summary:**

The paper tackles the task of converting RGBD indoor scans to 3D scenes for graphics rendering. It uses a multi-stage pipeline that first parses the scene into a structured scene graph, then performs 3D asset retrieval and material painting. The method achieves better quantitative performance on real-world scans than baselines.

**Questions:**

* What is the system's behavior when the retrieved asset departs far from the real object due to a retrieval error? How do the material painting and procedural reconstruction stages handle such discrepancies?

**Ethical Concerns:**

["NO or VERY MINOR ethics concerns only"]

**Final Justification:**

The rebuttal addresses most of my concerns. I agree with reviewer meLw that the proposed pipeline is engineering-heavy and the technical contribution over prior works is not significant. I can see it being practically useful, though, if as the authors argued, there is no prior work that works on the exact same setting. I raised my score as the rebuttal experiments resolved my concerns in the original review, but I'm open to hearing other reviewers' opinions.

**Limitations:**

Yes.

**Quality:**

2

**Strengths And Weaknesses:**

Strengths
* The paper presents a straightforward but well-motivated training-free pipeline to tackle an important task, and shows good empirical performance.

Weaknesses
* Errors may propagate across different stages in the pipeline. It would be informative for readers if in-depth failure analyses were presented.
* If I'm not missing it, the computation cost is promised in the checklist but not included in the supplementary.
* Some modules are added to the system to increase robustness, e.g., with VLM visual queries and renderings for analysis-by-synthesis for pose fitting. These modules should be thoroughly ablated and analyzed.

---

> ### Author Rebuttal · Authors · 2025-07-29
>
> We thank the reviewer for the thoughtful feedback and for acknowledging the importance of our task. Indeed, LiteReality is, to the best of our knowledge, the **first** method that reliably reconstructs indoor scenes into **compact CAD representations** with **realistic PBR materials** and **parsed layouts**. We respond to the weaknesses and questions as follows.
>
> **Weakness1: "Errors may propagate across different stages in the pipeline. It would be informative for readers if in-depth failure analyses were presented."**
>
> The error propagation is a natural challenge in multi-stage methods. To mitigate this, LiteReality is explicitly designed to achieve strong performance at each individual stage—demonstrated by our state-of-the-art retrieval similarity (Table 1) and best performance in object-centric PBR material estimation (Table 2, Figure 6, 7). Overall robustness is evidenced by high-quality reconstructions on both the ScanNet dataset and diverse real-world scenes (Table 3, Main Figure 5, Supp. Figure 1).
>
> To further analyze failure modes, we conducted a detailed user study based on four real-world scans captured using an iPhone—without any manual intervention. These reconstructions represent typical, challenging indoor environments. We evaluate each stage of the pipeline independently by asking users to assess the quality of outputs at each step.
>
> For each stage, users rated the results using the following scale:
> - Good: A faithful and realistic representation of the real-world object or layout.
> - Sufficient: Not an exact match, but still plausible in context (e.g., retrieving a three-door wardrobe for a two-door one that maintains overall scene realism).
> - Bad: Visibly inaccurate or implausible, breaking scene realism and requiring modification for acceptable results.
>
> **1. Scene Understanding Results**
>
> | Scene | Total | TP | FP | FN |
> |-|-|-|-|-|
> | Meeting Room | 21    | 21 | 0  | 1  |
> | Boardroom    | 48    | 48 | 1  | 0  |
> | Common Room  | 20    | 20 | 0  | 0  |
> | Study Room   | 12    | 12 | 0  | 0  |
>
> Failure cases include the window-side embedded seating area in the Boardroom, which was falsely detected as a sofa, and a chair in the Meeting Room that was not detected.
>
> **2. Retrieval Quality**
>
> | **Scene**      | **Total Objects** | **Good (%)** | **Sufficient (%)** | **Bad (%)** |
> |----------------|-------------------|--------------|---------------------|-------------|
> | Meeting Room   | 21                | 12 (57%)     | 7 (33%)             | 2 (10%)     |
> | Boardroom      | 48                | 40 (83%)     | 6 (13%)             | 2 (4%)      |
> | Common Room    | 20                | 16 (80%)     | 2 (10%)             | 2 (10%)     |
> | Study Room     | 12                | 6 (50%)      | 5 (42%)             | 1 (8%)      |
> | **Total**      | **101**           | **74 (73%)** | **20 (20%)**        | **7 (7%)**  |
>
> Typical retrieval errors include cases such as a round table being retrieved as a square table, and a glass-door cabinet retrieved as a glass shelf. Additional failure examples are shown in Supplementary Figure 9.
>
> **3. Material Quality**
>
> | Scene        | Total | Good (%)   | Sufficient (%) | Bad (%) |
> |--------------|-------|------------|----------------|---------|
> | Meeting Room | 21    | 19 (90%)   | 2 (10%)        | 0 (0%)  |
> | Boardroom    | 48    | 43 (90%)   | 4 (8%)         | 1 (2%)  |
> | Common Room  | 20    | 18 (90%)   | 2 (10%)        | 0 (0%)  |
> | Study Room   | 12    | 9 (75%)    | 2 (17%)        | 1 (8%)  |
> | **Total**    | 101   | 89 (88%)   | 10 (10%)       | 2 (2%)  |
>
> Overall, the material painting is robust. Occasional failures occur due to misidentified albedo colors or mismatches between segmentation and crop regions.
>
> **4. Combined Quality**
>
> | Scene        | Total | Good (%)   | Sufficient (%) | Bad (%) |
> |--------------|-------|------------|----------------|---------|
> | Meeting Room | 21    | 8 (38%)    | 12 (57%)       | 1 (5%)  |
> | Boardroom    | 48    | 39 (81%)   | 6 (13%)        | 3 (6%)  |
> | Common Room  | 20    | 11 (55%)   | 7 (35%)        | 2 (10%) |
> | Study Room   | 12    | 7 (58%)    | 4 (33%)        | 1 (8%)  |
> | **Total**    | 101   | 65 (64%)   | 29 (29%)       | 7 (7%)  |
>
> The combined quality reflects a few compounded errors, where imperfections in retrieval or material painting affect the final outcome. Nonetheless, the system remains robust across a diverse set of real-world scans. [see more in the video demo]
>
>
> **W2: the computation cost is promised in the checklist but not included in the supplementary**
>
> Our method runs on a single NVIDIA RTX 3090 GPU. Below is the runtime breakdown per scene:
>
> | Scene        | Preprocessing + Scene Parsing | Object Retrieval | Material Painting | Procedural Reconstruction |
> |--------------|-------------------------------|------------------|-------------------|----------------------------|
> | Boardroom    | 3m30s                          | 4m30s            | 51m               | 1m10s                      |
> | Common Room  | 1m20s                          | 3m39s            | 19m               | 48s                        |
> | Study Room   | 1m50s                          | 2m20s            | 14m               | 45s                        |
> | Meeting Room | 1m20s                          | 3m39s            | 19m               | 48s                        |
>
> These runtimes are measured on RTX 3090 GPU. We can further speed up the run time by increasing parallelization, especially during the material prediction stage.
>
> **W3: These modules should be thoroughly ablated and analyzed.**
>
> Below, we provide detailed ablation studies for both the **retrieval pipeline** and the **material Painting** module.
>
> **Retrieval Pipeline Ablation**
>
> | ID  | Semantic | Image Feat. | Pose R&C | MLLM | Avg. Chamfer (↓) |
> |-----|----------|-------------|----------|------|------------------|
> | I   | ✓        | ✓           |          |      | 0.142            |
> | II  | ✓        | ✓           | ✓        |      | 0.110            |
> | III | ✓        | ✓           |          | ✓    | 0.112            |
> | IV  | ✓        |             | ✓        |      | 0.104            |
> | V   | ✓        | ✓           | ✓        | ✓    | **0.097**        |
>
> Similar to Table 1, these experiments are conducted on Scan2CAD datasets, meaning retrieval similiarty using average Chamfer Distance. Each added module contributes meaningfully to improving performance.
>
> **Material Recovery Ablation**
>
> | ID  | Language | Visual | Albedo-only | RMSE (↓) | SSIM (↑) | LPIPS (↓) |
> |-----|----------|--------|-------------|-----------|-----------|------------|
> | I   | ✓        |        |             | 0.241     | 0.397     | 0.613      |
> | II  |          | ✓      |             | 0.322     | 0.239     | 0.658      |
> | III | ✓        | ✓      |             | 0.286     | 0.378     | 0.637      |
> | IV  | ✓        | ✓      | ✓           | **0.218** | **0.436** | **0.584**  |
>
> Previous methods often struggle to recover realistic materials under occlusion or imperfect lighting. In contrast, each component in our design contributes meaningfully to material quality, as evidenced by consistent improvements across RMSE, SSIM, and LPIPS metrics.
>
>
> **Q: What is the system's behavior when the retrieved asset departs far from the real object due to a retrieval error? How do material painting and reconstruction handle discrepancies?**
>
> We rigorously benchmarked on the Scan2CAD dataset and achieved SOTA results on retriveal. This shows that the likelihood of a poor retrieval is lower in our approach compared to existing methods. In the rare case where a retrieval error does occur, the system remains robust through the following mechanisms:
> 1. Robust Material Painting:
> The material recovery stage is **specifically designed to handle geometric misalignments** (Figure 6). Even with significant discrepancies between the retrieved object’s shape and the scanned image, LiteReality is able to transfer realistic appearance with PBR materials.
> 2. Physically Plausible Placement:
> The reconstruction pipeline ensures that all objects are still placed in physically plausible locations and poses, maintaining global scene consistency despite local retrieval errors.
>
> In cases of extreme mismatch, where the output may visibly diverge from reality, scene produced by our method are easily editable due to the decomposition of objects geometry and materails.
>
> Thank you again for your thoughtful review and do let us know if you have further questions.

---

> > ### Author Response · Authors · 2025-08-07
> >
> > Dear Reviewer,
> >
> > Thank you again for your time and effort in reviewing our paper. As the discussion period deadline is approaching, we’d be very grateful to hear your thoughts on our rebuttal—particularly whether it has addressed your concerns. We would also be happy to provide any further details or clarifications if needed.
> >
> > Best regards,
> > The LiteReality Team

---

> > > ### Comment · Reviewer_jpwk · 2025-08-08
> > > **Thank you for the rebuttals**
> > >
> > > I thank the authors for the additional explanations and experiments during the rebuttal.
> > >
> > > The additional experiments in the rebuttal are missing some important details, e.g., how many scenes the metrics are computed over.
> > >
> > > I agree with reviewer y37B that evaluation metrics such as object scale/orientation analysis in retrieval would help clarify the sources of errors in the full pipeline, even if this module in particular is not a claimed contribution. Additional runtime comparisons, as mentioned in the initial review from reviewer y37B, would also help provide the context and the performance-computation trade-off of the method compared to prior arts. The analysis and the runtime comparisons are still missing during the rebuttal.

---

> > > > ### Author Response · Authors · 2025-08-08
> > > >
> > > > Dear reviewer,
> > > >
> > > > Thanks for your additional question, please find the reply below.
> > > >
> > > > 1. Additioanl experimetn details.
> > > >
> > > > In the provided user study table above to showcase the error analysis, **each row represents one scene** that is captured with real-world scans. Specifically, their visual correspondences are:
> > > > - Boardroom (Figure 5, second row)
> > > > - Common Room (Figure 5, first row)
> > > > - Study Room (Figure 5, third row)
> > > > - Meeting Room (Figure 5, last row, right)
> > > >
> > > > Ablations are conducted using the same experimental settings as those in the paper. The Retrieval Pipeline Ablation, following Table 1, is tested on the entire ScanNet validation set, while the Material Recovery Ablation follows Table 2, using five scanned real-life scenes where the retrieval objects are pre-set for a controlled study (as agreed by Reviewer 5sYV, this is a sound design). These five scanned scenes and the pre-picked retrieval objects will also be released for future comparison, as stated in the paper.
> > > >
> > > > 2. Object Scale/Orientation Analysis
> > > >
> > > > We will perform bounding box labeling on the five real-world scenes to be released. This will allow us to quantitatively assess object scale, orientation, and completeness by comparing the reconstructed results against the ground truth. We plan to include this analysis in the final version of the paper and release the data to support this benchmarking.
> > > >
> > > > 3. Run time comparison.
> > > >
> > > > | Scene           | Method       | Preprocessing + Scene Parsing | Object Retrieval | Material Painting | Procedural Reconstruction |
> > > > |-----------------|--------------|-------------------------------|------------------|-------------------|----------------------------|
> > > > | **Boardroom**   | LiteReality  | 3m30s                         | 4m30s            | 51m               | 1m10s                      |
> > > > |                 | DC           | 3m30s                         | 2m               | –                 | 1m10s                      |
> > > > |                 | Phone2Proc   | 3m30s                         | 20s              | –                 | 1m10s                      |
> > > > | **Common Room** | LiteReality  | 1m20s                         | 3m39s            | 19m               | 48s                        |
> > > > |                 | DC           | 1m20s                         | 1m               | –                 | 48s                        |
> > > > |                 | Phone2Proc   | 1m20s                         | 10s              | –                 | 48s                        |
> > > > | **Study Room**  | LiteReality  | 1m50s                         | 2m20s            | 14m               | 45s                        |
> > > > |                 | DC           | 1m50s                         | 50s              | –                 | 45s                        |
> > > > |                 | Phone2Proc   | 1m50s                         | 10s              | –                 | 45s                        |
> > > > | **Meeting Room**| LiteReality  | 1m20s                         | 3m39s            | 19m               | 48s                        |
> > > > |                 | DC           | 1m20s                         | 20s              | –                 | 48s                        |
> > > > |                 | Phone2Proc   | 1m20s                         | 10s              | –                 | 48s                        |
> > > >
> > > > Please find the new table above for comparing runtime. We would like to highlight the following:
> > > > - DC and Phone2Proc cannot produce realistic reconstructions that adhere closely to real-world scans. They take less time for object retrieval—since retrieval similarity is not a key focus—and do not include any material painting stages.
> > > > - While DC and Phone2Proc emphasize diversity in reconstructed scenes for robust training, our method focuses on high-quality PBR material realism for faithful real-life reconstruction. We believe LiteReality is promising in generating high-quality synthetic data, with an additional graphics-ready layer (PBR materials) that is useful across many domains.
> > > >
> > > > Thank again for your reply and do let us know if you have further quesitons.

---

> > > > > ### Comment · Reviewer_jpwk · 2025-08-08
> > > > >
> > > > > Thank you for your rebuttal. I don't have further concerns and will raise my score.

---

> > > > > > ### Author Response · Authors · 2025-08-08
> > > > > >
> > > > > > Dear Reviewer,
> > > > > >
> > > > > > Thank you — we sincerely appreciate your thoughtful review and support of our work.
> > > > > >
> > > > > > Best regards,
> > > > > > The LiteReality Team

---

### Author Response · Authors · 2025-08-04

Dear Reviewers,

Thank you again for your thoughtful and constructive reviews of LiteReality. If you have any further questions or comments regarding our rebuttal or the method in general, we would be grateful to hear them and would be keen to engage further in discussion.

Best regards,
The LiteReality Team

---

### Author Response · Authors · 2025-08-09
**Summary of Rebuttal**

Dear Reviewers and Chairs,

Thank you so much for your effort in reviewing and handling our paper. Over the past weeks, we have had discussions with the reviewers, who have recognized the value of our work and provided valuable feedback for future improvements.

At the end of the discussion period, 2 out of 4 reviewers had submitted their mandatory acknowledgements and concluded the discussion. Both of these reviewers gave positive ratings:
- Reviewer Jpwk: **Agreed that we have addressed all concerns and raised the score to positive**.
- Reviewer y37B: **Stated that all the concerns have been addressed** and maintained a positive rating.

With the other two reviewers, who have not yet submitted the mandatory acknowledgement, we still had a round of discussion, and we appreciate their participation and feedback.

- Reviewer meLw: **Stated that all concerns have been addressed** and expressed ready to enter the next stage of discussion between AC-reviewers.
- Reviewer 5sYV: **Agreed that most of the original concerns have been addressed**, found our evaluation sound, and recognized that building a fully automated pipeline to robustly convert RGB-D scans into realistic CAD models with PBR materials is **novel and meaningful**. We have answered their clarifications and are awaiting their response.

### Summary
LiteReality is the first work capable of automatically converting messy RGB-D scans of real-world environments into a compact, structured, and realistic CAD representation with full PBR materials. The performance is tested on both real-world scans and the ScanNet dataset and shows robustness. We believe this tool will have impact for wide domain and be useful for the community [**all reviewers agreed upon this point**].
In the process of delivering this pipeline, we also pushed each individual component to its upper limit:
- Retrieval tasks – For selecting the best asset, we rigorously benchmarked our retrieval module performance on the Scan2CAD dataset and achieved SOTA results in retrieval similarity.
- Material recovery – We proposed a self-contained, lightweight PBR material recovery method that works well at scale, even under poor lighting and occlusion—conditions that are difficult for previous methods to handle.

Furthermore, to assist future development in this direction and to improve ease of use of this work.

- We built three rigorous benchmarks to evaluate individual components such as retrieval similarity, object-centric material recovery, and overall realism, supporting continued research in this area. Dataset used during evaluation will also be released. [**Reviewer 5sYV pointed out that our evaluation, which disentangles each step for control study, is sound.**]

- We provide a curated database containing manually cleaned and polished material segmentations for thousands of indoor furniture items, supporting retrieval- and material-painting–based object reconstruction.

Thank the reviewers and Chairs again for their effort and contributions to improving this work.

Best regards,
The LiteReality Team

---

### Decision · Program_Chairs · 2025-09-17

**Decision:**

Accept (poster)

**Comment:**

This paper was reviewed by four experts in the field, and all of them are inclined to recommend acceptance of the paper after the rebuttal phase. The reviewers appreciate that the novel and important problem setting: producing realistic CAD representations from raw RGB-D scans at room scale, with reasonable technical improvements and relatively comprehensive experiments. Thus, this paper achieves the NeurIPS acceptance requirement.